# Improving Pretraining Data Using Perplexity Correlations

**Tristan Thrush, Christopher Potts & Tatsunori Hashimoto**

Department of Computer Science

Stanford University

Stanford, CA 94305, USA

{tthrush,cgpotts,thashim}@stanford.edu

## Abstract

Quality pretraining data is often seen as the key to high-performance language models. However, progress in understanding pretraining data has been slow due to the costly pretraining runs required for data selection experiments. We present a framework that avoids these costs and selects high-quality pretraining data without *any* LLM training of our own. Our work is based on a simple observation: LLM losses on many pretraining texts are correlated with downstream benchmark performance, and selecting high-correlation documents is an effective pretraining data selection method. We build a new statistical framework for data selection centered around estimates of perplexity-benchmark correlations and perform data selection using a sample of 90 LLMs taken from the Open LLM Leaderboard on texts from tens of thousands of web domains. In controlled pretraining experiments at the 160M parameter scale on 8 benchmarks, our approach outperforms DSIR on every benchmark, while matching the best data selector found in DataComp-LM, a hand-engineered bigram classifier. We have now also updated this paper to include results from preregistered experiments with new pretraining data on an aggregation of 22 benchmarks up to the 1.4B scale, showing increasing improvements of our method over others with more scale. A pip package with full documentation can be found here: https://github.com/TristanThrush/perplexity-correlations.

## 1 Introduction

Dataset curation is increasingly crucial for training high-quality large language models (LLMs). As pretraining datasets have grown, from under 200B tokens in 2020 (Raffel et al., 2020; Gao et al., 2020) to 240T tokens today (Li et al., 2024), it has become critical to identify subsets of the available data that will lead to the best LLMs, and a wide range of methods have arisen to meet these needs (Ilyas et al., 2022; Xie et al., 2023a;b; Engstrom et al., 2024; Everaert & Potts, 2024; Liu et al., 2024; Llama Team, 2024). However, data-driven approaches to data selection typically involve expensive model retraining steps that limit their effectiveness, and no algorithm has been reported to consistently beat or match hand-crafted classifiers for data selection (Li et al., 2024).

Is training new LLMs necessary for data selection? Instead of training our own models, can we use the growing collection of publicly available, high-performance LLMs (Wolf et al., 2019; Beeching et al., 2023) to perform data valuation and selection? This would have significant benefits: we could leverage the millions of dollars collectively spent on building these LLMs, and we would have coverage over a large, heterogeneous collection of high-performance models varying in size, architectures, and pretraining data distribution.

Despite these advantages, using existing models for pretraining data selection is challenging, as the training data for these models are often unknown and heterogeneous. Our key observation is that data selection can be done using two observable features of *all* public models today: 1) all open-weight models produce a causal language modeling loss for a given text, and 2) all of them can be

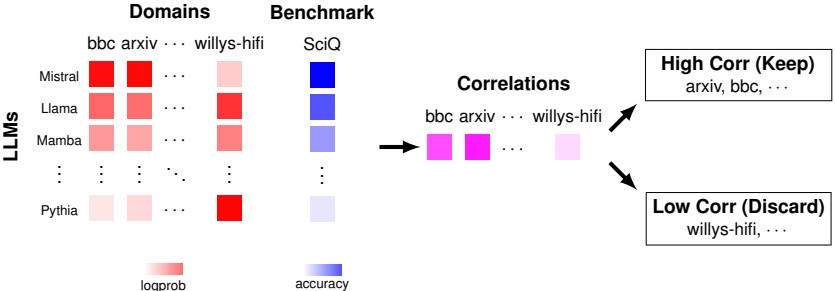

Figure 1: We pretrain on domains where lower loss is generally correlated with higher downstream performance. Our approach does this by taking public, pretrained LLMs and measuring correlations across their log-likelihoods (left, red matrix) and performance on a target benchmark (center, blue vector). We then perform data selection by training a fastText classifier that distinguishes high correlation domains from others. This approach is on par with the best-known data selection methods in our experiments, despite requiring no human selection of high-quality domains.

evaluated on benchmarks. Prior work has found systematic relationships between web corpus loss and benchmark performance (Wei et al., 2022; Huang et al., 2024), which suggests the possibility of using correlations between perplexity and benchmark scores as the basis for a data selection policy.

In the present paper, we pursue this possibility and find a radically simple approach that is also effective: we select data via *perplexity correlations* (Figure 1), where we select data domains (e.g. wikipedia.org, stackoverflow.com, etc.) for which LLM log-probabilities are highly correlated with downstream benchmark performance. To enable our approach, we complement our algorithm with a statistical framework for correlation-based data selection and derive correlation estimators that perform well over our heterogeneous collection of LLMs.

We validate our approach using a collection of pretrained causal LLMs on the Hugging Face Open LLM Leaderboard (Beeching et al., 2023) and find that perplexity correlations are predictive of an LLM's benchmark performance. Importantly, we find that these relationships are robust enough to enable reliable data selection that targets downstream benchmarks. In controlled pretraining experiments at the 160M parameter scale on eight benchmarks, our approach strongly outperforms DSIR (Xie et al., 2023b) (a popular training-free data selection approach based on n-gram statistics) while generally matching the performance of the best method validated at scale by Li et al. (the OH-2.5 +ELI5 fastText classifier; Joulin et al. 2016) without any parameter tuning or human curation. In followup experiments at the 160M to 1.4B parameter scale which we pre-registered, our approach outperforms the best Li et al. filter on the main benchmark from their paper (an aggregate of 22 benchmarks) when filtering from their base data pool, and both approaches remain close to each other when filtering from their extensively pre-filtered pool. We further find that the performance of our approach strengthens with increasing scale.

## 2 RELATED WORK

To go beyond the status quo of deduplication, perplexity filtering, and hand-curation (Laurençon et al., 2022; BigScience, 2023; Marion et al., 2023; Abbas et al., 2023; Groeneveld et al., 2024; Soldaini et al., 2024; Penedo et al., 2024; Llama Team, 2024), targeted methods have been proposed to filter pretraining data so that the resulting LLM will achieve higher scores on given benchmarks. There are lightweight approaches that use n-gram overlap (Xie et al., 2023b) or embedding similarity (Everaert & Potts, 2024) to select training data that is similar to data from a given benchmark. There are also less-scalable methods that require training proxy LLMs on different data mixtures (Ilyas et al., 2022; Xie et al., 2023a; Engstrom et al., 2024; Liu et al., 2024; Llama Team, 2024).

Given the high costs of proxy-based data selection methods, they have primarily been used to select among human-curated pretraining data mixtures (Llama Team, 2024; Li et al., 2024) rather than a high dimensional space of mixtures. Our work takes an orthogonal approach and builds upon recent observational studies that have found scaling relationships that hold across collections of uncontrolled and diverse LLMs (Owen, 2024; Ruan et al., 2024). While these studies do not examine

loss-to-performance relationships or derive useful data selection methods from them, we know that losses and performance are generally highly correlated. Validation losses on samples of text corpora are commonly used as a proxy for downstream performance when comparing LLMs pretrained on the same data distribution (Kaplan et al., 2020; Hoffmann et al., 2022; Wei et al., 2022), even if they have different architectures (Poli et al., 2023; Peng et al., 2023; Gu & Dao, 2024).

According to a recent survey of data selection approaches by Li et al. (2024), the heavier-weight pretraining data selection methods have not shown large gains, and the current state-of-the-art across many tasks is primitive: a fixed fastText classifier (Joulin et al., 2016) combined with an English filter as a final layer after extensive deduplication and filtering. Are we missing important information that we can efficiently extract from a diverse collection of already trained models, larger and more diverse than any single organization is likely to produce? We show evidence supporting this hypothesis – simple loss-performance correlation coefficients are effective when used for data selection.

## 3    PROBLEM SETTING

Our goal is to build predictive models of how pretraining data distributions affect downstream benchmark performance and use them to build better language models. Unfortunately, this task is challenging and computationally expensive. A standard approach adopted in paradigms such as datamodeling (Ilyas et al., 2022) is to obtain $N$ different pretraining distributions $\{\mathbf{p}_i : i \in [N], \mathbf{p}_i \in \mathbb{R}_0^{+D}\}$ over $D \gg N$ domains (e.g. arxiv.org, stackoverflow.com, etc.), pretrain and measure model errors on a target benchmark $y_i \in [0, 1]$, and fit a model $\mathbf{p} \to y$. This approach requires $N$ LLM training runs, performed at a scale sufficient to obtain non-random performance on $y$. This can cost tens to hundreds of millions of dollars for hard benchmarks such as MMLU, where even the performance of 1B parameter LLMs often does not exceed random chance (Beeching et al., 2023).

Instead, our work considers the following *observational* setting that requires no training. We obtain $N$ pretrained, high-performance LLMs that vary in pretraining data, tokenizer, architecture, and scale (i.e. models on Hugging Face's OpenLLM leaderboard). If we could train a predictor $\mathbf{p} \to y$ on these $N$ models, we could avoid large scale model training. Unfortunately, this is impossible as the training data for these models is often proprietary, and so we have no knowledge of $\mathbf{p}$.

The key observation of our work is that we can replace $p_{i,j}$ (the unobserved sampling probability of model $i$'s data selection policy on document $j$) with an observable surrogate $x_{i,j}$, which is the negative log-likelihood of document $j$ under model $i$.[1] We can then build a regression model that relates negative log-likelihood $\mathbf{x}_i$ and benchmark error $y_i$. Using this model, we can select pretraining data from domains $j$ for which decreasing the loss $x_{i,j}$ is predicted to rapidly decrease error $y_i$.

**The perplexity-performance hypothesis.** We formulate the task of predicting errors $y_i$ from negative log-probabilities $\mathbf{x}_i$ as a single-index model (SIM),

$$y_i = f(\langle \boldsymbol{\theta}^*, \mathbf{x}_i \rangle + \epsilon_i) \tag{1}$$

where $f : \mathbb{R} \mapsto \mathbb{R}$ is some unknown monotonically increasing univariate function, $\epsilon_i$ is zero-mean noise which is independent of $\mathbf{x}$, and $\boldsymbol{\theta}^* \in \mathbb{R}^D$ are unknown weights over $D$ domains.

A single index model is highly flexible (due to the arbitrary, monotone $f$) and has the advantage that we do not need to estimate the nonlinear function $f$ if our goal is to optimize model performance. We can see this directly from the monotonicity of $f$ as

$$\langle \boldsymbol{\theta}^*, \mathbf{x}_i \rangle + \epsilon_i < \langle \boldsymbol{\theta}^*, \mathbf{x}_j \rangle + \epsilon_j \iff f(\langle \boldsymbol{\theta}^*, \mathbf{x}_i \rangle + \epsilon_i) < f(\langle \boldsymbol{\theta}^*, \mathbf{x}_j \rangle + \epsilon_j). \tag{2}$$

**Data selection from perplexity correlations.** The weights $\boldsymbol{\theta}^*$ tell us which domain perplexities are correlated with downstream performance. However, this isn't sufficient for data selection. Even if we know how model likelihoods relate to model performance, we do not know how data selection affects likelihoods. Even worse, this data mixture to likelihood relationship *cannot* be learned observationally, as we do not know the data mixture of any of our models.

---

[1]To be precise, we use bits-per-byte, which normalizes the sequence negative log-likelihood with the number of UTF-8 bytes. This is defined in terms of the length of the string in tokens $L_T$, the length of the string in UTF-8 bytes $L_B$, and the cross entropy loss $\ell$ as BPB $= \frac{L_T \ell}{L_B \ln(2)}$

Despite this, we show that there is a clean approach for optimizing the data mixture. Our core observation is the following: *if we find a nonnegative $\boldsymbol{\theta}^*$, sampling proportional to $\boldsymbol{\theta}^*$ is always a good choice.* More formally, we see that this sampling distribution defines the pretraining loss such that optimizing the training loss directly optimizes the downstream task via the single index model.

**Proposition 1** *Suppose that $\boldsymbol{\theta}^*$ weights are non-negative. Then, for models with associated likelihoods $\mathbf{x} \in \mathcal{X} \subset \mathbb{R}^D$, the minimizer of the pretraining loss over the $\boldsymbol{\theta}^*$ sampling distribution $\mathbb{E}_{j \sim \boldsymbol{\theta}^*}[x_j]$ also has the lowest expected downstream error according to the single index model:*

$$\arg\min_{\mathbf{x} \in \mathcal{X}} \mathbb{E}_{j \sim \boldsymbol{\theta}^*}[x_j] = \arg\min_{\mathbf{x} \in \mathcal{X}} \mathbb{E}[f(\langle \boldsymbol{\theta}^*, \mathbf{x} \rangle + \epsilon)].$$

This observation follows directly from the fact that we can normalize any non-negative $\boldsymbol{\theta}^*$ into a distribution (and shift the normalization constant into $f$) which allows us to write the inner product in the single-index model as a monotone function of the expected pretraining loss:

$$y = f(\langle \boldsymbol{\theta}^*, \mathbf{x} \rangle + \epsilon) = f(\mathbb{E}_{j \sim \boldsymbol{\theta}^*}[x_j] + \epsilon). \tag{3}$$

Proposition 1 allows us to entirely avoid the task of finding the optimal data mixture for a target likelihood. Instead, we pick sampling distributions that make the pretraining loss a monotone function of the predicted downstream error. Afterward, we can rely on our ability to optimize the loss to optimize downstream performance.

This view gives us a straightforward roadmap for data selection in the remainder of the paper: estimate a set of domains where loss and downstream benchmark performance is highly correlated, and then constrain our $\boldsymbol{\theta}^*$ estimates to be a pretraining data sampling distribution.

## 4 METHODS

We now describe the details of our approach, starting by presenting the algorithm itself and the intuitions behind it, followed by a more precise and mathematical justification for the various steps.

### 4.1 ALGORITHM

**Estimating $\boldsymbol{\theta}^*$.** The parameter $\theta_j^*$ measures the relationship between log-likelihoods in domain $j$ and downstream performance. Because of this, we might naturally expect $\theta_j^*$ to be related to nonlinear correlation coefficients between $x$ and $y$. Our work uses a simple correlation measure,

$$\gamma_j = \sum_{\substack{1 \le k,l \le n \\ k \ne l}} \operatorname{sign}(y_k - y_l)(\operatorname{rank}_j(x_{k,j}) - \operatorname{rank}_j(x_{l,j}))$$

where $\operatorname{rank}_j(x)$ is the rank of $x$ among $\{x_{1,j} \ldots x_{N,j}\}$. This formula is intuitive: when model $k$ does better than model $l$, what percentile is model $k$'s log-likelihood compared to model $l$'s? The functional form also has the benefit of being a principled estimate of $\boldsymbol{\theta}^*$. In particular, we show in sections below that in expectation, the ranking of domains in $\boldsymbol{\gamma}$ exactly matches that of $\boldsymbol{\theta}^*$ (under standard high-dimensional regression assumptions; see Section 4.2 for a complete discussion). We note, though, that this is not the only correlation coefficient that performs well in our perplexity correlations toolkit (see Appendix H) and we prove that Spearman's rank correlation (Spearman, 1904) is principled in the same way (see Appendix C).

**Selecting pretraining data.** Suppose that we have an accurate estimate $\gamma_j$ which is nonnegative. In this case, we could use $\gamma_j$ directly as a data selection procedure and Proposition 1 would ensure that minimizing the population pretraining loss minimizes downstream errors. Unfortunately, $\gamma_j$ can be negative and the finite number of tokens per domain can make it difficult to minimize the population pretraining loss. Thus, we must project $\gamma_j$ onto the set of reasonable pretraining data distributions that are nonnegative and account for the per-domain token counts.

What is a good way to project a set of domain rankings estimated via $\boldsymbol{\gamma}$ into a pretraining sampling distribution? Intuitively, if wikipedia.org has a $\gamma_j = 0.5$ and arxiv.org is $\gamma_k = 0.9$, it would be natural to select tokens in order of $\boldsymbol{\gamma}$, preferring tokens from arxiv.org over tokens from wikipedia.org.

Having established the ordering of domains, the remaining question is how many tokens we take for each domain. We follow recent observations that repeating data degrades performance (Abbas et al.,

2023) to arrive at a simple selection algorithm: select domains in greatest to least $\boldsymbol{\gamma}$, taking all the tokens in each domain once, until we exhaust our total pretraining token budget.

**Full algorithm.** Together, these steps result in a simple, parameter-free algorithm that calculates our rank correlation coefficient, and selects domains in order from largest to smallest coefficient. We show this process explicitly with pseudocode in Algorithm 1 (see Appendix A), and additionally show an extra step where we train a fastText (Joulin et al., 2016) classifier (using standard settings and bigram features from Li et al. 2024) which distinguishes our selected documents and domains from the rest of the pool. The fastText classifier allows us to perform data selection at a single-page level, and scale the selection process to larger datasets. We also found the classifier to slightly improve downstream performance over directly selecting the documents. More information on the specifics of the data selection approaches that we tested is given in Appendix G.

## 4.2 THEORY

We now study the approach closely and show that our choices for the correlation coefficient and projection step are extensions of the classic, high-dimensional single index model estimator of Plan et al. (2016). We describe the basic single-index model estimators first, describe our extensions, and then conclude with a discussion on how our estimator and results deviate from the theory. A discussion of other potential estimation paradigms is provided in Appendix E.

### 4.2.1 HIGH-DIMENSIONAL ESTIMATION OF SINGLE INDEX MODELS

For our theory, we consider the standard high-dimensional regression setting of Plan et al. (2016) and Chen & Banerjee (2017). Here, our goal is to estimate the unknown weights $\boldsymbol{\theta}^*$ in a single-index model $y_i = f(\langle \boldsymbol{\theta}^*, \mathbf{x}_i \rangle + \epsilon_i)$, with $\mathbf{x}_i \sim \mathcal{N}(\mathbf{0}, \mathbf{I})$ for $\|\boldsymbol{\theta}^*\|_2 = 1$ (assumed without loss of generality, as $\|\boldsymbol{\theta}^*\|_2$ can be absorbed by $f$).

Our starting point is the classic result of Plan et al. (2016), who showed

$$\mathbb{E}\left[y_k \mathbf{x}_k\right] = c\boldsymbol{\theta}^*, \tag{4}$$

for some positive constant $c$ and $1 \le k \le N$. Closely related is the result of Chen & Banerjee (2017), who developed a robust estimator quite similar to ours,

$$\mathbb{E}\left[\text{sign}(y_k - y_l)(\mathbf{x}_k - \mathbf{x}_l)\right] = \beta\boldsymbol{\theta}^* \tag{5}$$

for any $1 \le k, l \le N$ (where $k \ne l$) and some positive constant $\beta$. Both of these results clearly identify that for the high-dimensional single-index model in the Gaussian setting, generalized correlation coefficients provide consistent estimates of the true regression coefficient $\boldsymbol{\theta}^*$.

### 4.2.2 DERIVING OUR ESTIMATOR

Both Plan et al. and Chen & Banerjee provide moment-matching style estimators that consistently recover $\boldsymbol{\theta}^*$ in high-dimensional, sparse settings. However, we found that both estimators directly use the values of $x$, and this resulted in brittle estimates due to outliers in language model log-likelihoods. While outlier removal is one possibility, we found that a simpler approach was to robustify the estimator of Chen & Banerjee (2017) to outliers in $x$.

Recall that our estimate $\boldsymbol{\gamma}$ is a U-statistic, defined as pairwise sums of

$$\text{sign}(y_i - y_j)(\Phi(\mathbf{x}_i) - \Phi(\mathbf{x}_j)), \tag{6}$$

for any $1 \le i, j \le N$ (where $i \ne j$), where $\Phi$ is the empirical CDF (effectively, normalized ranks) of the $\mathbf{x}$ values. This estimate is significantly less sensitive to outliers than that of Chen & Banerjee (2017), as the empirical CDF is bounded between zero and one.

We study this estimate theoretically in the Gaussian setting, where we consider the asymptotically equivalent estimator with $\Phi$ as the CDF of the standard Gaussian. In this case, we can show that this modified estimator is also consistent in recovering $\boldsymbol{\theta}^*$.

**Theorem 1** *When $\epsilon \sim \mathcal{N}(0, \sigma^2)$, we have:*

$$\mathbb{E}[\text{sign}(y_i - y_j)(\Phi(\mathbf{x}_i) - \Phi(\mathbf{x}_j))] = \frac{2}{\pi} \sin^{-1}\left(\frac{\boldsymbol{\theta}^*}{2\sqrt{1 + \sigma^2}}\right). \tag{7}$$

The proof is in Appendix B. Because we assume $||\boldsymbol{\theta}^*||_2 = 1$ and the expected value in Equation 7 must be in $[-1, 1]$, we are always in the domain of $\sin^{-1}$ and can invert it. After inverting, we get:

$$\hat{\boldsymbol{\theta}} \propto \sin\left(\frac{\pi}{2}\mathbb{E}\left[\operatorname{sign}(y_i - y_j)(\Phi(\mathbf{x}_i) - \Phi(\mathbf{x}_j))\right]\right) \tag{8}$$

as an estimate for $\boldsymbol{\theta}^*$, where the constant $2\sqrt{1 + \sigma^2}$ term due to noise has been dropped.

Beyond the fact that our estimator is consistent, we can show an even tighter connection to the Chen & Banerjee estimator: our estimates agree when running the original estimator on rank-transformed data. More specifically, for two models $\mathbf{x}_i$ and $\mathbf{x}_j$ with the estimated model rankings $\langle \hat{\boldsymbol{\theta}}, \mathbf{x}_i \rangle > \langle \hat{\boldsymbol{\theta}}, \mathbf{x}_j \rangle$, the expected ranking under rank-transformation (i.e. $\Phi(\mathbf{x})$) matches this ranking.

**Corollary 1** *Suppose that $\hat{\boldsymbol{\theta}}$ is any vector of fixed weights and $\mathbf{x} \sim \mathcal{N}(\mathbf{0}, \mathbf{I})$. Then, conditioning on the event $\langle \hat{\boldsymbol{\theta}}, \mathbf{x}_i \rangle < \langle \hat{\boldsymbol{\theta}}, \mathbf{x}_j \rangle$, we have with probability 1 that:*

$$\langle \hat{\boldsymbol{\theta}}, \mathbb{E}[\Phi(\mathbf{x}_i) \mid \langle \hat{\boldsymbol{\theta}}, \mathbf{x}_i \rangle < \langle \hat{\boldsymbol{\theta}}, \mathbf{x}_j \rangle]\rangle < \langle \hat{\boldsymbol{\theta}}, \mathbb{E}[\Phi(\mathbf{x}_j) \mid \langle \hat{\boldsymbol{\theta}}, \mathbf{x}_i \rangle < \langle \hat{\boldsymbol{\theta}}, \mathbf{x}_j \rangle]\rangle. \tag{9}$$

This proof follows from the same calculations as Theorem 1 and is given in Appendix B.

### 4.2.3 SELECTING DATA FOR PRETRAINING

Recall that our algorithm for data selection is to constrain $\boldsymbol{\gamma}$ to be a valid sampling distribution and then sample directly from this estimate. For now, we focus on constraining $\hat{\boldsymbol{\theta}}$, and we will see at the end of this section that we can apply the same constraint to $\boldsymbol{\gamma}$ directly to get the same result. The theory of constrained estimation for $\hat{\boldsymbol{\theta}}$ is simple and well-understood, with both Plan et al. (2016) and Chen & Banerjee (2017) extensively studying the problem of estimating $\hat{\boldsymbol{\theta}}$ under a known convex constraint set $C$. In particular, Plan et al. (2016) show that performing a $L_2$ projection via $\hat{\boldsymbol{\theta}}^{\text{proj}} = \arg\min_{\boldsymbol{\theta} \in C} \|\boldsymbol{\theta} - \hat{\boldsymbol{\theta}}\|_2$ provides improved convergence rates that depend on the Gaussian mean width of $C$ rather than the ambient dimension, and Chen & Banerjee (2017) show similar results when maximizing the linear correlation $\hat{\boldsymbol{\theta}}^{\text{proj}} = \arg\min_{\boldsymbol{\theta} \in C \subseteq B_D} -\langle \boldsymbol{\theta}, \hat{\boldsymbol{\theta}} \rangle$.

We take a similar approach here. We define a convex constraint set $C$ that forces $\hat{\boldsymbol{\theta}}$ to be a reasonable sampling distribution and find the best sampling distribution via the linear correlation approach.

We define $C$ as the combination of two sets of constraints. First, we must have a valid sampling distribution, so we constrain $\hat{\boldsymbol{\theta}}$ to lie in the simplex. As we noted above, it is well-known that duplicating data harms performance (Abbas et al., 2023), so we constrain $\hat{\boldsymbol{\theta}}$ to avoid this by limiting the maximum weight on domains: if we will pretrain on $m$ tokens overall, we enforce $\theta_i^* \leq \tau_i, \forall i \in [1, D]$, where $\tau_i$ is set so $\tau_i m$ is the number of tokens from the $i$-th domain accessible for training.

The resulting linear program has a simple solution and takes the form of initializing $\hat{\boldsymbol{\theta}}^{\text{proj}}$ to $\mathbf{0}$ and then iterating through the values in $\hat{\boldsymbol{\theta}}$ from largest to smallest, setting the value at the corresponding index of $\hat{\boldsymbol{\theta}}^{\text{proj}}$ to the maximum allowable value, until $\hat{\boldsymbol{\theta}}^{\text{proj}}$ sums to 1 (see Appendix D for a proof).

**Theorem 2** *Suppose we want to solve:*

$$\hat{\boldsymbol{\theta}}^{\text{proj}} = \arg\min_{\boldsymbol{\theta} \in \mathbb{R}^D} -\langle \boldsymbol{\theta}, \hat{\boldsymbol{\theta}} \rangle,$$

*subject to:*

$$\sum_{i=1}^{D} \theta_i = 1$$

$$0 \leq \theta_i \leq \tau_i, \forall i \in [1, D],$$

*where $\tau_i > 0$ are fixed values. Then, the solution is:*

$$\hat{\theta}_k^{\text{proj}} = \begin{cases} \tau_k & \text{if } \sum_{j: r_j(\hat{\theta}_j) \geq r_k(\hat{\theta}_k)} \tau_j \leq 1 \\ 1 - \sum_{j: r_j(\hat{\theta}_j) > r_k(\hat{\theta}_k)} \tau_j & \text{if } \sum_{j: r_j(\hat{\theta}_j) \geq r_k(\hat{\theta}_k)} \tau_j \geq 1 \wedge \sum_{j: r_j(\hat{\theta}_j) > r_k(\hat{\theta}_k)} \tau_j \leq 1 \\ 0 & \text{otherwise} \end{cases} \tag{10}$$

*where $r$ breaks ties between $\hat{\theta}_j$ and $\hat{\theta}_k$ for $k \neq j$; otherwise $r$ keeps ordinal relationships the same.*

We note that while the use of this linear program is in line with the constrained estimators proposed in Chen & Banerjee (2017), the $L_2$ projection is arguably more natural, and does not require assuming that $\|\hat{\boldsymbol{\theta}}\|_2 = 1$ for asymptotic recovery conditions. We derive similar closed-form expressions for this quadratic case in Appendix D, but do not use this approach for two separate reasons. First, the $L_2$ projection depends on the $L_2$ norm of $\hat{\boldsymbol{\theta}}$, unlike the linear program which only depends on the ranks of the values in $\hat{\boldsymbol{\theta}}$. The challenge with determining the norm is that the exact recovery result in Equation (7) requires knowledge of the noise level, and the trigonometric functions rely strongly on the Gaussian structure of $x$. Because of this, we are unlikely to be able to estimate the norm of $\hat{\boldsymbol{\theta}}$ with any accuracy, and the only way to avoid this would be to treat the norm as a hyperparameter, which adds unnecessary complexity. The second reason is empirical (although possibly a consequence of the first) – we found that the linear projection performed better across a wide range of benchmarks and conditions (see Appendix H).

We conclude by relating our theory to the full algorithm in Section 4.1. The estimation step for $\gamma$ is the finite sample, U-estimate of the expectation in Equation (8), dropping the nonlinear transform $\sin$ and $\pi/2$ as these two terms do not change the rankings of the domains. The data selection step directly applies our projection in Equation (10), and we make use of the fact that this projection only relies on rankings among the domains to use $\gamma$ rather than an exact estimate for $\boldsymbol{\theta}^*$.

## 5 RESULTS

We first pretrain 160M-parameter LLMs from scratch to study our primary goal of selecting pretraining data to improve downstream performance, and then we present a brief overview of results on experiments which we preregistered in an earlier version of this document (most experiments show a trend of even greater improvements at larger scales – up to 1.4B parameters – on new benchmarks and data pools; more detail is in Appendices N and M). Finally, we present an analysis on the ability of losses to predict downstream performance. Throughout our experiments, we use single-index models trained using Algorithm 1. As shown in the algorithm, we train the fastText classifier on selected vs. unselected domains and use the classifier to filter the pretraining data at the page-level.

**Input data matrix X.** To build the input data matrix, **X**, we collected byte normalized loss values from a sample of 90 Open LLM Leaderboard (Beeching et al., 2023) LLMs that we could run without errors. Concretely, these values are defined as bits-per-byte $\frac{L_T \ell}{L_B \ln(2)}$ where $L_T$ is the token count, $L_B$ is the number of UTF-8 bytes, and $\ell$ is the per-token cross-entropy (Gao et al., 2020). For our initial experiments, we collected these values on the "sample" subset[2] of the RedPajama V2 (RPJv2) dataset (Together Computer, 2023) for all domains with $\geq 25$ pages in the sample. There are 9,841 domains/features. Specifics are in Appendix F. The data for the additional preregistered experiments is discussed in Appendix N. A detailed principal components analysis of **X**, which reveals a variety of salient embedded information in the losses, is in Appendix L.

**Target benchmark performance** $y$. For our initial experiments, we constructed a target vector, **y**, for LAMBADA (Paperno et al., 2016), ARC Easy (Clark et al., 2018), PIQA (Bisk et al., 2020), and SciQ (Welbl et al., 2017). These are all of the tasks reported in the Pythia scaling experiments for which a model in the 160M parameter range could meaningfully perform above chance. We also constructed target vectors for LAMBADA$_{IT}$, LAMBADA$_{FR}$, LAMBADA$_{DE}$, and LAMBADA$_{ES}$, which are subsets of LAMBADA translated into Italian, French, German, and Spanish by Black (2023). These languages match those in RPJv2, where each page is conveniently tagged as one of five languages: English, Spanish, French, German, and Italian. The correspondence between our target benchmark languages and the RPJv2 metadata allows us to easily include language filtering baselines. For the preregistered experiments, we use 22 more benchmarks (see Appendix N).

### 5.1 INITIAL PRETRAINING EXPERIMENTS

We begin by validating our algorithm in the end-to-end task of pretraining data selection with controlled experiments at the 160M parameter, 3.2B token, scale. The low compute requirements of this setting allow us to more extensively study replicates and ablations in Appendix H within the timeframe of a few days. While 160M models are small, this is far from an easy setting for our data

---

[2]https://huggingface.co/datasets/togethercomputer/RedPajama-Data-V2

Table 1: Average rankings of data selection methods (lower is better) for 8 evals. Correlation-based filtering beats baselines by a wide margin, and matches the current best open data filter from Li et al. (2024). Our approach significantly beats the default filter in Li et al. (2024) and loses slightly after additional manual language filtering that depends on the target task (+ manual Lang Filter).

| Method | None | Lang Filt | DSIR (Xie et al., 2023b) | Handcrafted fastText + EN Lang Filter (Li et al., 2024) | Handcrafted fastText w/o Lang Filter | Handcrafted fastText + manual Lang Filter | Perplexity Correlations |
|---|---|---|---|---|---|---|---|
| Avg. Rank | 3.750 | 4.000 | 4.500 | 3.750 | 3.250 | 1.375 | 1.750 |

selection algorithm. Most of the Open LLM Leaderboard models are 10 to $100\times$ larger than the 160M scale, and our single index model must extrapolate substantially from $\approx$7B scale models to our small-scale validation setting (see Appendix K for a histogram of model sizes).

**Pretraining data and setting.** For the initial pretraining experiments, we used the "sample-100B" subset of RPJv2. This is much larger than the sample that we used to compute our estimate. We filtered this data so it contains only the domains used for our estimate, and then tokenized the data with the Pythia tokenizer. The vast majority of the domains from our BPB matrix were present in this larger sample of text. However, 42 (out of 9,841) were not, and so we removed them from our estimate. For every data selection method that we tested, the task was to further select 3.2B tokens for pretraining, which is Chinchilla-optimal (Hoffmann et al., 2022) for the 160M-parameter LLM.

**Baselines.** We compare against several baseline data-selection methods. First, we present the results of uniformly sampling from the available pretraining data. Then we use the language tags present in RPJv2 to filter only for the language matching the target task. In addition to these commonsense baselines, we also run DSIR (Xie et al., 2023b), a lightweight training data selection technique based on n-gram overlaps that Li et al. (2024) found to be competitive with proxy LLM-based techniques and was also validated at scale (Parmar et al., 2024). Finally, we run the state-of-the-art method for pretraining data quality filtering found by Li et al., which is a fastText classifier that beats all of the heavier-weight proxy-LLM methods tested. The classifier was trained on a benchmark-agnostic and handcrafted objective, which is to classify data as Common Crawl[3] (low quality) or OH2.5 (Teknium, 2023) and Reddit ELI5 (Fan et al., 2019) (high quality). It is combined with an English filter in Li et al.; we present results for this fastText filter with and without the English filter.

**Model and hyperparameters.** We use the Pythia 160M LLM configuration from Biderman et al. (2023) and optimize the hyperparameters, including learning rate, weight decay, and warmup, to minimize loss on the uniform sampling (no selection algorithm) baseline. Training hyperparameters were fixed across all methods. We provide additional training and evaluation details in Appendix G.

**Results.** We report average rankings over initial benchmarks in Table 1, and we find that our approach significantly outperforms the basic baselines of random sampling, language filtering, and DSIR. Compared to the existing state of the art from Li et al. (2024), our approach beats the performance of the default, English-filtered fastText classifier, but loses slightly once we add in a manual language filtering step to enable better performance on the multilingual LAMBADA datasets. For the maintext comparisons, we use the optional fastText classifier from our algorithm to select pretraining data at the page levels, but we show ablations without the classifier in Appendix H.

Figure 2 shows how each data selection method affects benchmark performance in more detail. Each block of rows represents a data selection method, while an individual row represents an LLM within a method that targets a particular benchmark or set of benchmarks. Columns represent benchmarks. We see that language filtering and perplexity correlations both clearly optimize for the target benchmark: within each block, the benchmark column matching each row typically performs best. The pattern is much less obvious for DSIR – the heatmap looks more uniform across LLMs with different task targets. We also see that while language filtering has significant impacts on model performance, our performance significantly exceeds the impact of language filtering across all tested benchmarks. For some more analysis, including the types of domains selected, see Appendix O and J.

Finally, we note that our results are somewhat insensitive to the specifics of the perplexity-correlation procedure we present in Algorithm 1. We show in Appendix H that varying the projection method (linear, $L_2$) and even using Spearman rank correlations (Spearman, 1904) often work better than the

---

[3]https://commoncrawl.org

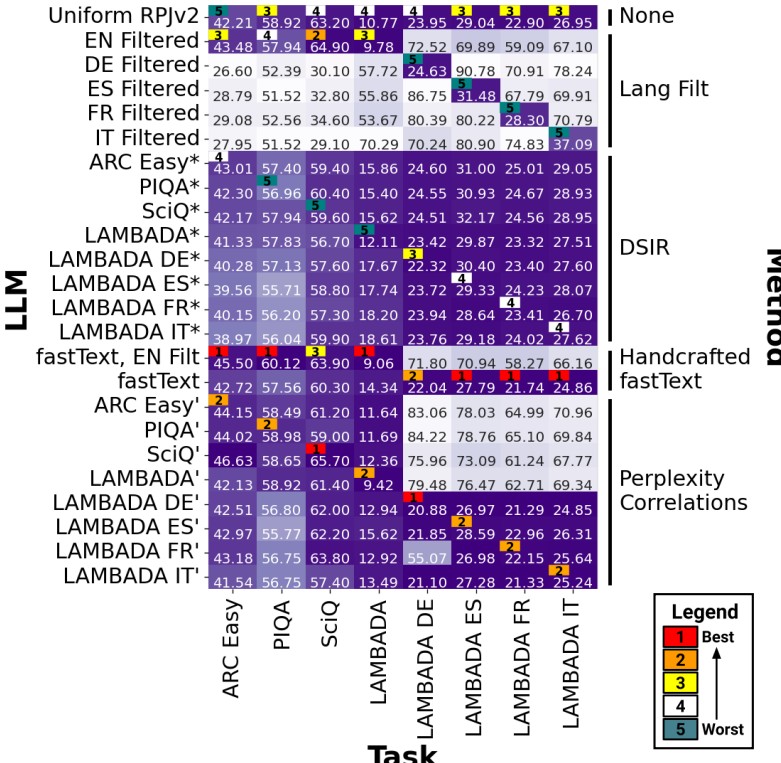

Figure 2: Pretraining results with different data selection methods. Each row is an LLM, and each column is a task. The number in the upper left indicates the ranking of the method when targeting that benchmark compared to other methods (lower is better). Numbers within the heatmap denote accuracy for all benchmarks except the LAMBADA tasks for which the values are log perplexities (where lower scores are better). We find that our approach appropriately optimizes data mixes for the target language and benchmark, and matches the fastText baseline across most benchmarks.

baselines. This suggests that the performance of our approach is not dependent on the precise form of the estimator, but holds broadly across perplexity-correlation relationships; we may also be able to prove the Appendix B and C results for many other rank correlation coefficients. Additionally, our approach performs better with the optional fastText classifier that our algorithm trains, possibly because it operates at the page-level instead of the domain-level.

## 5.2 PREREGISTERED PRETRAINING EXPERIMENTS

For our preregistered experiments, we target aggregates of many different benchmarks: "DCLM Core", which is an aggregate of 22 benchmarks, and "Non-EN LAMBADA", which is an aggregate of 4. We pretrain on DCLM (Li et al., 2024) data pools at chinchilla optimal (Hoffmann et al., 2022) levels going up to 1.4B parameters, and we generally see increasing perplexity correlations performance with scale. See Figure 3 and Appendix N.

## 5.3 PERFORMANCE RANK PREDICTIONS

Comparing model performance rankings predicted by our regression to the ground truth, we find generally accurate predictions. Figure 4 shows 5-fold leave-out plots for PIQA, and LAMBADA$_{FR}$ with rank predictions given by $\langle \hat{\boldsymbol{\theta}}^{\text{proj}}, \Phi(\mathbf{x}) \rangle$. Every point in the plot is a held-out point: we estimated $\theta^*$ five times, holding out a different 20% of the data each time, and plotted the held-out predictions. We find that our estimator achieves high ordinal prediction performance across all target tasks. We include 5-fold leave-out $R^2$ scores for all tasks in Appendix I.

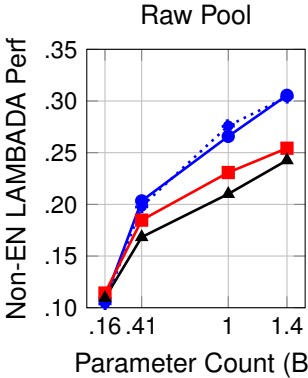
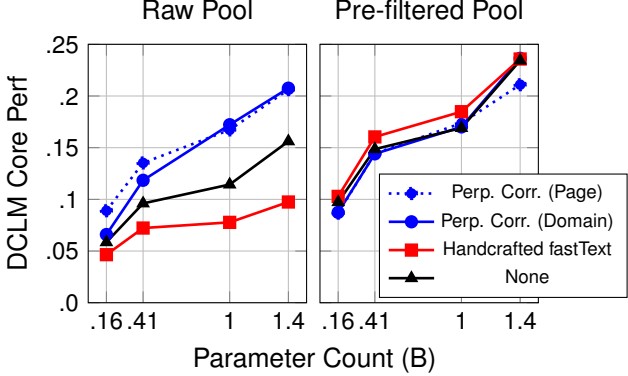

Figure 3: Preregistered experiment results. We did not see a benefit from using perplexity correlations when the dataset is already extensively filtered, but saw large consistent benefits otherwise, with the benefits increasing with scale. For the pre-filtered pool, the largest correlation coefficient was .33 and the smallest was .23 with the vast majority of domains being over .29, so we could have predicted no or small gains before pretraining. In the raw pool for DCLM Core, the largest coefficient was .32 and the smallest was −.07. Pre-filtered pool results for Non-EN LAMBADA are not shown because there is only English in the pre-filtered pool. See Appendix N for more details.

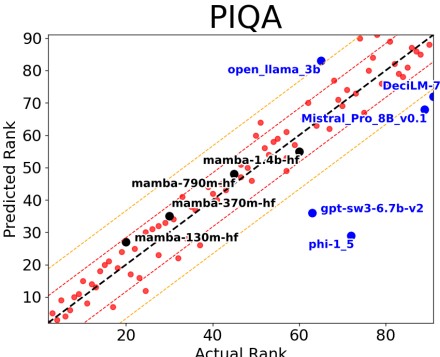
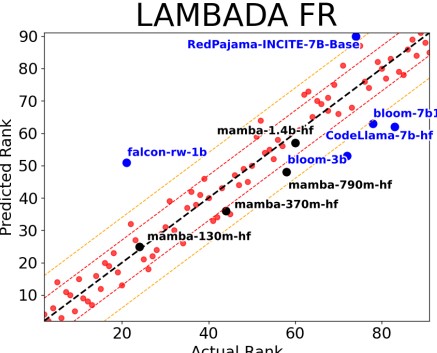

Figure 4: Rank predictions given by $\langle \hat{\boldsymbol{\theta}}^{\mathrm{proj}}, \Phi(\mathbf{x}) \rangle$ for PIQA and LAMBADA FR. A standard deviation ($\sigma$) from the ideal fit is shown in red. $2\sigma$ is shown in orange. Many models outside $2\sigma$ (shown in blue) are trained on atypical data such as multilingual data, code, or GPT-4 (Brown et al., 2020) outputs. Models with atypical architectures (i.e. Mamba; Gu & Dao 2024) are shown in black. Generally, our estimate tightly predicts ordinal benchmark performance from web corpus losses.

Finally, we discuss outliers in our prediction of model performance. Our predictions are accurate for LLMs with unusual architectures (e.g. Mamba; Gu & Dao 2024), and the smallest/largest vocabulary sizes, context sizes, and parameter sizes. However, LLMs that were trained on unusual data are not as well predicted by our approach (e.g. Phi; Gunasekar et al. 2023). We may require a bigger or more diverse pretraining data pool or set of models to find estimates that work well for these models.

## 6 CONCLUSION

Does high-performance data selection require hand-crafted heuristics or prohibitively expensive model training runs? Our work demonstrates an alternative, viable approach: leveraging existing, public models as a source of information for data selection. Pretraining experiments suggest that a simple, correlation-based approach to selecting data can be effective, but more broadly, we show how to 1) use single-index models as surrogates for downstream performance and 2) build models that relate *losses* to downstream performance and use the surrogates effectively to select data.

ACKNOWLEDGMENTS

We thank Jack Spilecki for conversations on the mathematical aspects of the work. We also thank Zitong Yang, Yangjun Ruan, and Lisa Li for their helpful feedback throughout the project, Ludwig Schmidt and Samir Gadre for discussions on scaling laws involving benchmark perplexity, Rohan Pandey for conversations about scaling laws, Sung Min Park for discussions on drafts of this work, and William Held for conversations about data selection. This work is supported in part by a grant from Sandia National Laboratories, and gifts from Open Philanthropy, Meta, Panasonic Research, and the Tianqiao and Chrissy Chen Institute. Any opinions, findings, and conclusions or recommendations expressed in this material are those of the authors and do not necessarily reflect the views of Sandia National Laboratories. Tristan Thrush is supported in part by the Stanford Graduate Fellowship.

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

## A  MAIN ALGORITHM

---

**Algorithm 1** Perplexity Correlation Based Data Selection

---

**Input:** Benchmark error vector $\mathbf{y} \in [0,1]^N$, log-loss matrix normalized as bits-per-byte $\mathbf{X} \in \mathbb{R}_0^{+N \times D}$, available tokens per domain $\mathbf{a} \in \mathbb{N}^D$, and pretraining token target $b \in \mathbb{N}$.
**Output:** Target token counts per domain $\mathbf{t} \in \mathbb{N}_0^D$, a fastText classifier to filter pretraining data.
**Initialize:** $\boldsymbol{\gamma} \leftarrow \mathbf{0} \in \mathbb{R}^D, \mathbf{t} \leftarrow [0\ldots] \in \mathbb{N}_0^D$, counter $\leftarrow 0$.
$\mathbf{r}_0, \mathbf{r}_1, \ldots, \mathbf{r}_N \leftarrow \text{rank}(\mathbf{x}_0, \mathbf{x}_1, \ldots, \mathbf{x}_N)$       ▷ 1. Compute the $\boldsymbol{\gamma}$ correlation coefficient
**for** $i, j \in 0$ **to** $N$ **do**
    $\boldsymbol{\gamma} \leftarrow \boldsymbol{\gamma} + \text{sign}(y_i - y_j) \cdot (\mathbf{r}_i - \mathbf{r}_j)$
**for** $i \in$**ArgSort**$(\boldsymbol{\gamma}$, descending=True$)$ **do**       ▷ 2. Select most corr. domains (linear projection)
    $t_i \leftarrow \min(a_i, b - \text{counter})$
    counter $\leftarrow$ counter $+ a_i$
    **if** counter $\geq b$ **then**
        **Break**
classifier = trainFastText(positive $= 1_{t>0}$, negative $= 1_{t=0}$)
**Return** $\mathbf{t}$, classifier

---

## B  ESTIMATOR SOLUTION

### B.1  LEMMA 1

**Statement of Lemma 1** *Define the PDF of* HalfNormal *as* $f(x; \sigma) = \frac{\sqrt{2}}{\sigma\sqrt{\pi}} e^{-\frac{x^2}{2\sigma^2}}$ *for $x > 0$ and $0$ otherwise. Now, suppose:*

- *$\boldsymbol{\beta}$ is a vector with $||\boldsymbol{\beta}||_2 = 1$*

- *$\mathbf{Z}_1, \mathbf{Z}_2$ are vectors $\sim \mathcal{N}(\mathbf{0}, \mathbf{I})$*

- *$\epsilon \sim \mathcal{N}(0, \sigma^2)$*

- *$Z' \sim \mathcal{N}(0, 1)$*

- *$Z_+ \sim$ HalfNormal$(1)$.*

*Then we have:*

$$Z_{1j} | \langle \mathbf{Z}_1 - \mathbf{Z}_2, \boldsymbol{\beta} \rangle + \epsilon > 0 \stackrel{d}{=} Z'\sqrt{1 - \frac{\beta_j^2}{2 + \sigma^2}} + \frac{\beta_j}{\sqrt{2 + \sigma^2}} Z_+,$$

*where $Z_{1j}$ is the $j$-th entry of $\mathbf{Z}_1$.*

*Proof:* First, note:

$$Z_{1j}|\langle \mathbf{Z}_1 - \mathbf{Z}_2, \boldsymbol{\beta}\rangle + \epsilon > 0 \stackrel{d}{=} Z_{1j}\Bigg| \left\langle \begin{bmatrix} \mathbf{Z}_1 \\ \mathbf{Z}_2 \\ \epsilon/\sigma \end{bmatrix}, \begin{bmatrix} \boldsymbol{\beta} \\ -\boldsymbol{\beta} \\ \sigma \end{bmatrix} \right\rangle > 0 \stackrel{d}{=} Z_{1j}\Bigg| \left\langle \begin{bmatrix} \mathbf{Z}_1 \\ \mathbf{Z}_2 \\ \epsilon/\sigma \end{bmatrix}, \begin{bmatrix} \boldsymbol{\beta} \\ -\boldsymbol{\beta} \\ \sigma \end{bmatrix} / \sqrt{2 + \sigma^2} \right\rangle > 0,$$

where $\begin{bmatrix} \cdot \\ \cdot \\ \cdot \end{bmatrix}$ denotes the vector-valued result of concatenating vectors and scalars. For readability, we

set $\mathbf{Z}_c = \begin{bmatrix} \mathbf{Z}_1 \\ \mathbf{Z}_2 \\ \epsilon/\sigma \end{bmatrix}$ and $\boldsymbol{\beta}_c = \begin{bmatrix} \boldsymbol{\beta} \\ -\boldsymbol{\beta} \\ \sigma \end{bmatrix} / \sqrt{2 + \sigma^2}$.

Given that $\boldsymbol{\beta}_c$ is unit-norm (by supposition, $\boldsymbol{\beta}$ is unit-norm), and every element of $\mathbf{Z}_c$ is $\sim \mathcal{N}(0,1)$ (even $\epsilon/\sigma$), we can easily split a conditional random vector containing $Z_{1j}$ into a conditionally dependent component and independent component:

$$\mathbf{Z}_c | \langle \mathbf{Z}_c, \boldsymbol{\beta}_c \rangle > 0 \overset{d}{=} (\mathbf{I} - \boldsymbol{\beta}_c \boldsymbol{\beta}_c^\top) \mathbf{Z}'' + \boldsymbol{\beta}_c \mathbf{Z}_+.$$

The first term is orthogonal to $\boldsymbol{\beta}_c$ and so it is the part of $\mathbf{Z}_c$ that is not subject to the condition. In the unconditional case, $\mathbf{Z}_c \sim \mathcal{N}(\mathbf{0}, \mathbf{I})$ and so $\mathbf{Z}'' \sim \mathcal{N}(\mathbf{0}, \mathbf{I})$. The second term is the part of $\mathbf{Z}_c$ that is in the direction of $\boldsymbol{\beta}_c$. $\mathbf{Z}_+ \sim \text{HalfNormal}(\mathbf{I})$ because our dot product condition is satisfied for half of the possible non-orthogonal $\mathbf{Z}_c$ values. Now, we focus on finding $\mathbf{Z}_c | \langle \mathbf{Z}_c, \boldsymbol{\beta}_c \rangle > 0$ for a single index $j$. We have (for $C$ defined to be the dimensionality of $\boldsymbol{\beta}_c$):

$$((\mathbf{I} - \boldsymbol{\beta}_c \boldsymbol{\beta}_c^\top) \mathbf{Z}'')_j + (\boldsymbol{\beta}_c \mathbf{Z}_+)_j = Z_j''(1 - \beta_{cj}^2) - \sum_{\substack{1 \le i \le C \\ i \ne j}} Z_i'' \beta_{cj} \beta_{ci} + \beta_j Z_{+j}$$

Now, note that $Z_j''(1 - \beta_{cj}^2) - \sum_{\substack{1 \le i \le C \\ i \ne j}} Z_i'' \beta_{cj} \beta_{ci}$ is the sum of independent zero-mean Gaussians, so it itself is a zero-mean Gaussian with variance:

$$(1 - \beta_{cj}^2)^2 + \sum_{\substack{1 \le i \le C \\ i \ne j}} \beta_{cj}^2 \beta_{ci}^2 = 1 - 2\beta_{cj}^2 + \beta_{cj}^4 + \sum_{\substack{1 \le i \le C \\ i \ne j}} \beta_{cj}^2 \beta_{ci}^2$$

$$= 1 - 2\beta_{cj}^2 + \beta_{cj}^2 \sum_{1 \le i \le C} \beta_{ci}^2$$

$$= 1 - \beta_{cj}^2,$$

where we simplified the expression by recalling that $\boldsymbol{\beta}_c$ is unit norm. So we have that $Z_{1j}$ is:

$$Z' \sqrt{1 - \beta_{cj}^2} + \beta_{cj} Z_+ = Z' \sqrt{1 - \frac{\beta_j^2}{2 + \sigma^2}} + \frac{\beta_j}{\sqrt{2 + \sigma^2}} Z_+,$$

for $Z' \sim \mathcal{N}(0,1)$. As a corollary, we can see that $Z_{2j}$ under the same condition is given by:

$$Z' \sqrt{1 - \frac{\beta_j^2}{2 + \sigma^2}} + \frac{-\beta_j}{\sqrt{2 + \sigma^2}} Z_+.$$

## B.2 LEMMA 2

**Statement of Lemma 2** *Suppose that $\Phi$ is the CDF of a standard Gaussian, $a$ and $c$ are constants, and $Z \sim \mathcal{N}(0,1)$. Then we have:*

$$\mathbb{E}[\Phi(aZ + c)] = \Phi\left(\frac{c}{\sqrt{1 + a^2}}\right).$$

*Proof:* By the definition of the CDF of a standard Gaussian, we have:

$$\mathbb{E}[\Phi(aZ + c)] = \mathbb{E}[P(X \le aZ + c)],$$

where $X \sim \mathcal{N}(0,1)$. Continuing, we have:

$$= \mathbb{E}[P(X - aZ - c \le 0)].$$

Now, note that $X - aZ - c$ is the sum of independent Gaussian random variables with given mean and variance; it itself is a Gaussian random variable $\sim \mathcal{N}(-c, a^2 + 1)$. To find $P(X - aZ - c \le 0)$, we can evaluate its CDF at 0:

$$= \mathbb{E}\left[\Phi\left(\frac{c}{\sqrt{a^2 + 1}}\right)\right] = \Phi\left(\frac{c}{\sqrt{a^2 + 1}}\right).$$

### B.3  LEMMA 3

**Statement of Lemma 3** *Suppose $\Phi$ is the standard Gaussian CDF, $Z_+ \sim \text{HalfNormal}(1)$, and $b$ and $a$ are constants. Then we have:*

$$\mathbb{E}\left[\Phi\left(\frac{Z_+ b}{\sqrt{a^2+1}}\right)\right] = \frac{1}{2} + \frac{1}{\pi}\tan^{-1}\left(\frac{b}{\sqrt{a^2+1}}\right).$$

*Proof:* By the definition of expected value, we can take the following integral where $f_{Z_+}$ is the PDF of $Z_+$. We integrate from $0$ instead of $-\infty$ because the PDF of the Standard Half Normal is $0$ in the domain below $0$:

$$\mathbb{E}\left[\Phi\left(\frac{Z_+ b}{\sqrt{a^2+1}}\right)\right] = \int_0^\infty \Phi\left(\frac{zb}{\sqrt{a^2+1}}\right) f_{Z_+}(z) dz$$

$$= \int_0^\infty \Phi\left(\frac{zb}{\sqrt{a^2+1}}\right) \frac{\sqrt{2}}{\sqrt{\pi}} e^{\frac{-z^2}{2}} dz$$

$$= \frac{1}{\sqrt{2\pi}}\left(\int_0^\infty e^{\frac{-z^2}{2}} dz + \int_0^\infty \text{erf}\left(\frac{zb}{\sqrt{2}\sqrt{a^2+1}}\right) e^{\frac{-z^2}{2}} dz\right) \quad (*).$$

The second integral is generally non-trivial to solve, but luckily we can solve it by using Equation 2 in Section 4.3 of the integral table from Ng & Geller (1968), which states:

$$\int_0^\infty \text{erf}(cx) e^{-d^2 x^2} dx = \frac{\sqrt{\pi}}{2d} - \frac{1}{d\sqrt{\pi}}\tan^{-1}\left(\frac{d}{c}\right)$$

Where $c$ and $d$ are real and positive. We split the solution by cases: $b > 0$, $b = 0$, and $b < 0$. We find that in every case, we can manipulate our integral so that the solution is trivial or the constant inside the $\text{erf}(\cdot)$ is positive (and so we can use the integral table). In every case, we find that the solution is $\frac{1}{2} + \frac{1}{\pi}\tan^{-1}\left(\frac{b}{\sqrt{a^2+1}}\right)$.

**Case 1:** $b > 0$. We can use the integral table directly:

$$(*) = \frac{1}{\sqrt{2\pi}}\left(\frac{\sqrt{\pi}}{\sqrt{2}} + \frac{\sqrt{\pi}}{\sqrt{2}} - \frac{\sqrt{2}}{\sqrt{\pi}}\tan^{-1}\left(\frac{\sqrt{a^2+1}}{b}\right)\right)$$

$$= \frac{1}{2} + \frac{1}{2} - \frac{1}{\pi}\tan^{-1}\left(\frac{\sqrt{a^2+1}}{b}\right).$$

Then, using the identity:

$$\tan^{-1} x + \tan^{-1}\frac{1}{x} = \frac{\pi}{2} \text{ if } x > 0,$$

we find the following:

$$= \frac{1}{2} + \frac{1}{\pi}\tan^{-1}\left(\frac{b}{\sqrt{a^2+1}}\right).$$

**Case 2:** $b = 0$. Note that $\text{erf}(0) = 0$; we do not have to use the integral table:

$$(*) = \frac{1}{\sqrt{2\pi}}\left(\frac{\sqrt{\pi}}{\sqrt{2}} + 0\right)$$

$$= \frac{1}{2}.$$

Because $\tan^{-1}(0) = 0$, we have:

$$= \frac{1}{2} + \frac{1}{\pi}\tan^{-1}\left(\frac{b}{\sqrt{a^2+1}}\right).$$

**Case 3:** $b < 0$. Because $\text{erf}(\cdot)$ is an odd function, we can pull the negative out:

$$(*) = \frac{1}{\sqrt{2\pi}}\left(\int_0^\infty e^{\frac{-z^2}{2}} dz - \int_0^\infty \text{erf}\left(\frac{z|b|}{\sqrt{2}\sqrt{a^2+1}}\right) e^{\frac{-z^2}{2}} dz\right).$$

Now we can use the integral table as in the $b > 0$ case:

$$= \frac{1}{\sqrt{2\pi}} \left( \frac{\sqrt{\pi}}{\sqrt{2}} - \frac{\sqrt{\pi}}{\sqrt{2}} + \frac{\sqrt{2}}{\sqrt{\pi}} \tan^{-1} \left( \frac{\sqrt{a^2+1}}{|b|} \right) \right)$$

$$= \frac{1}{2} + \frac{1}{2} - \frac{1}{\pi} \tan^{-1} \left( \frac{\sqrt{a^2+1}}{|b|} \right).$$

We can then use the same identity again:

$$\tan^{-1} x + \tan^{-1} \frac{1}{x} = \frac{\pi}{2} \text{ if } x > 0$$

to get:

$$= \frac{1}{2} - \frac{1}{\pi} \tan^{-1} \left( \frac{|b|}{\sqrt{a^2+1}} \right).$$

Because $\tan^{-1}$ is an odd function, we can put the negative inside of it:

$$= \frac{1}{2} + \frac{1}{\pi} \tan^{-1} \left( \frac{b}{\sqrt{a^2+1}} \right).$$

### B.4 FULL PROOF

Here, we prove:

$$\mathbb{E}[\text{sign}(y_1 - y_2)(\Phi(\mathbf{x}_1) - \Phi(\mathbf{x}_2))] = \frac{2}{\pi} \sin^{-1} \left( \frac{\boldsymbol{\theta}^*}{\sqrt{4 + 2\sigma_1^2 + 2\sigma_2^2}} \right)$$

with $y_1, y_2, \Phi(\mathbf{x}_1), \Phi(\mathbf{x}_2)$, and $\boldsymbol{\theta}^*$ defined in the main text, for the case where $\epsilon_1$ and $\epsilon_2$ are zero-mean Gaussian noise $\sim \mathcal{N}(0, \sigma_1^2)$ and $\sim \mathcal{N}(0, \sigma_2^2)$, respectively.

It is easy to see that this is a more general version of the following theorem.

**Theorem 1** *When $\epsilon \sim \mathcal{N}(0, \sigma^2)$, we have:*

$$\mathbb{E}[\text{sign}(y_i - y_j)(\Phi(\mathbf{x}_i) - \Phi(\mathbf{x}_j))] = \frac{2}{\pi} \sin^{-1} \left( \frac{\boldsymbol{\theta}^*}{2\sqrt{1 + \sigma^2}} \right). \tag{7}$$

*Proof:* By symmetry, we have:

$$\mathbb{E}[\text{sign}(y_1 - y_2)(\Phi(\mathbf{x}_1) - \Phi(\mathbf{x}_2))]$$

$$= \frac{1}{2}\mathbb{E}[\Phi(\mathbf{x}_1) - \Phi(\mathbf{x}_2) | \text{sign}(y_1 - y_2) > 0] + \frac{1}{2}\mathbb{E}[-(\Phi(\mathbf{x}_1) - \Phi(\mathbf{x}_2)) | \text{sign}(y_1 - y_2) < 0].$$

By increasing monotonicity of $f$, we have $\text{sign}(y_1 - y_2) > 0 \iff \langle \mathbf{x}_1 - \mathbf{x}_2, \boldsymbol{\theta}^* \rangle + \epsilon_\Delta > 0$, for $\epsilon_\Delta = \epsilon_1 - \epsilon_2 \sim \mathcal{N}(0, \sigma_1^2 + \sigma_2^2)$. So:

$$= \frac{1}{2}\mathbb{E}[\Phi(\mathbf{x}_1) - \Phi(\mathbf{x}_2) | \langle \mathbf{x}_1 - \mathbf{x}_2, \boldsymbol{\theta}^* \rangle + \epsilon_\Delta > 0]$$

$$+ \frac{1}{2}\mathbb{E}[-(\Phi(\mathbf{x}_1) - \Phi(\mathbf{x}_2)) | \langle \mathbf{x}_1 - \mathbf{x}_2, \boldsymbol{\theta}^* \rangle + \epsilon_\Delta < 0].$$

Because $\mathbf{x}_1 \overset{d}{=} \mathbf{x}_2$ and $\epsilon_\Delta \overset{d}{=} -\epsilon_\Delta$, the two expected values above are the same:

$$= \mathbb{E}[\Phi(\mathbf{x}_1) - \Phi(\mathbf{x}_2) | \langle \mathbf{x}_1 - \mathbf{x}_2, \boldsymbol{\theta}^* \rangle + \epsilon_\Delta > 0].$$

By linearity of expectation:

$$= \mathbb{E}[\Phi(\mathbf{x}_1) | \langle \mathbf{x}_1 - \mathbf{x}_2, \boldsymbol{\theta}^* \rangle + \epsilon_\Delta > 0] - \mathbb{E}[\Phi(\mathbf{x}_2) | \langle \mathbf{x}_1 - \mathbf{x}_2, \boldsymbol{\theta}^* \rangle + \epsilon_\Delta > 0].$$

Now, we focus on finding the overall estimate for a single index $j$. By Lemma 1, we have, for $Z \sim \mathcal{N}(0, 1)$ and $Z_+ \sim \text{HalfNormal}(1)$:

$$\Phi(x_{1j}) | \langle \mathbf{x}_1 - \mathbf{x}_2, \boldsymbol{\theta}^* \rangle + \epsilon_\Delta > 0 \overset{d}{=} \Phi(Za + Z_+ b_1).$$

Here, $a = \sqrt{1 - \frac{(\theta_j^*)^2}{2+\sigma_1^2+\sigma_2^2}}$ and $b_1 = \frac{\theta_j^*}{\sqrt{2+\sigma_1^2+\sigma_2^2}}$. As a corollary of Lemma 1, we can see:

$$\Phi(x_{2j})|\langle \mathbf{x}_1 - \mathbf{x}_2, \boldsymbol{\theta}^* \rangle + \epsilon_\Delta > 0 \overset{d}{=} \Phi(Za + Z_+ b_2).$$

Where $b_2 = -\frac{\theta_j^*}{\sqrt{2+\sigma_1^2+\sigma_2^2}}$. So for the index $j$, our estimate is:

$$\mathbb{E}[\Phi(Za + Z_+ b_1)] - \mathbb{E}[\Phi(Za + Z_+ b_2)]$$
$$= \mathbb{E}[\mathbb{E}[\Phi(Za + c)|c = Z_+ b_1]] - \mathbb{E}[\mathbb{E}[\Phi(Za + c)|c = Z_+ b_2]].$$

Using Lemma 2, we have:

$$= \mathbb{E}\left[\Phi\left(\frac{Z_+ b_1}{\sqrt{a^2 + 1}}\right)\right] - \mathbb{E}\left[\Phi\left(\frac{Z_+ b_2}{\sqrt{a^2 + 1}}\right)\right].$$

Then, using Lemma 3, we have:

$$= \frac{1}{2} + \frac{1}{\pi} \tan^{-1}\left(\frac{b_1}{\sqrt{a^2 + 1}}\right) - \frac{1}{2} - \frac{1}{\pi} \tan^{-1}\left(\frac{b_2}{\sqrt{a^2 + 1}}\right)$$
$$= \frac{1}{\pi} \tan^{-1}\left(\frac{b_1}{\sqrt{a^2 + 1}}\right) - \frac{1}{\pi} \tan^{-1}\left(\frac{b_2}{\sqrt{a^2 + 1}}\right).$$

Using the fact that $\tan^{-1}$ is an odd function and $b_2 = -b_1$, we get:

$$= \frac{2}{\pi} \tan^{-1}\left(\frac{b_1}{\sqrt{a^2 + 1}}\right).$$

Now, we write $a$ and $b_1$ in terms of $\theta_j^*$:

$$= \frac{2}{\pi} \tan^{-1}\left(\frac{\frac{\theta_j^*}{\sqrt{2+\sigma_1^2+\sigma_2^2}}}{\sqrt{2 - \frac{(\theta_j^*)^2}{2+\sigma_1^2+\sigma_2^2}}}\right)$$

$$= \frac{2}{\pi} \tan^{-1}\left(\frac{\frac{\theta_j^*}{\sqrt{4+2\sigma_1^2+2\sigma_2^2}}}{\sqrt{1 - \left(\frac{\theta_j^*}{\sqrt{4+2\sigma_1^2+2\sigma_2^2}}\right)^2}}\right).$$

Using the identity $\sin^{-1} x = \tan^{-1}\left(\frac{x}{\sqrt{1-x^2}}\right)$, we have:

$$= \frac{2}{\pi} \sin^{-1}\left(\frac{\theta_j^*}{\sqrt{4 + 2\sigma_1^2 + 2\sigma_2^2}}\right).$$

### B.5 COROLLARY 1

**Corollary 1** *Suppose that $\hat{\boldsymbol{\theta}}$ is any vector of fixed weights and $\mathbf{x} \sim \mathcal{N}(\mathbf{0}, \mathbf{I})$. Then, conditioning on the event $\langle \hat{\boldsymbol{\theta}}, \mathbf{x}_i \rangle < \langle \hat{\boldsymbol{\theta}}, \mathbf{x}_j \rangle$, we have with probability 1 that:*

$$\langle \hat{\boldsymbol{\theta}}, \mathbb{E}[\Phi(\mathbf{x}_i) \mid \langle \hat{\boldsymbol{\theta}}, \mathbf{x}_i \rangle < \langle \hat{\boldsymbol{\theta}}, \mathbf{x}_j \rangle] \rangle < \langle \hat{\boldsymbol{\theta}}, \mathbb{E}[\Phi(\mathbf{x}_j) \mid \langle \hat{\boldsymbol{\theta}}, \mathbf{x}_i \rangle < \langle \hat{\boldsymbol{\theta}}, \mathbf{x}_j \rangle] \rangle. \tag{9}$$

To see this, we can find:

$$\mathbb{E}[\Phi(\mathbf{x}_1) - \Phi(\mathbf{x}_2)|\langle \hat{\boldsymbol{\theta}}, \mathbf{x}_1 \rangle + \epsilon_1 > \langle \hat{\boldsymbol{\theta}}, \mathbf{x}_2 \rangle + \epsilon_2] = \mathbb{E}[\Phi(\mathbf{x}_1) - \Phi(\mathbf{x}_2)|\langle \hat{\boldsymbol{\theta}}, \mathbf{x}_1 - \mathbf{x}_2 \rangle + \epsilon_\Delta > 0]$$

Note that we have already computed this expected value in the proof above; for an index $j$, it is:

$$\frac{2}{\pi} \sin^{-1}\left(\frac{\hat{\theta}_j}{\sqrt{4 + 2\sigma_1^2 + 2\sigma_2^2}}\right).$$

Because $\sin^{-1}$ is an odd function, the above expression has the same sign as $\hat{\theta}_j$. Because the values at every index of $\mathbb{E}[\Phi(\mathbf{x}_1) - \Phi(\mathbf{x}_2)]$ under our condition and $\hat{\boldsymbol{\theta}}$ are the same sign, we have $\langle \mathbb{E}[\Phi(\mathbf{x}_1) - \Phi(\mathbf{x}_2)], \hat{\boldsymbol{\theta}} \rangle > 0$, so $\langle \hat{\boldsymbol{\theta}}, \mathbb{E}[\Phi(\mathbf{x}_1)] \rangle > \langle \hat{\boldsymbol{\theta}}, \mathbb{E}[\Phi(\mathbf{x}_2)] \rangle$.

## C SPEARMAN RANK ESTIMATOR SOLUTION

### C.1 LEMMA 4

**Statement of Lemma 4** *Suppose $\Phi$ is the standard Gaussian CDF, $\mathbf{Z}$ is a vector $\sim \mathcal{N}(\mathbf{0}, \mathbf{I})$, $\epsilon \sim N(0, \sigma^2)$, $\boldsymbol{\beta}$ is a vector with $\|\boldsymbol{\beta}\|_2 = 1$, and $a$ is a constant. Then we have:*

$$\mathbb{E}[\Phi(Z_j)|\langle \boldsymbol{\beta}, \mathbf{Z}\rangle + \epsilon = a] = \Phi\left(\frac{\beta_j a}{(1+\sigma^2)\sqrt{2 - \frac{\beta_j^2}{1+\sigma^2}}}\right)$$

*Proof:* Note that

$$Z_j|(\langle \mathbf{Z}, \boldsymbol{\beta}\rangle + \epsilon = a) \overset{d}{=} Z_j\left|\left(\left\langle \begin{bmatrix} \mathbf{Z} \\ \epsilon/\sigma \end{bmatrix}, \begin{bmatrix} \boldsymbol{\beta} \\ \sigma \end{bmatrix}\right\rangle = a\right)\right.$$

$$\overset{d}{=} Z_j\left|\left(\left\langle \begin{bmatrix} \mathbf{Z} \\ \epsilon/\sigma \end{bmatrix}, \begin{bmatrix} \boldsymbol{\beta} \\ \sigma \end{bmatrix}/\sqrt{1+\sigma^2}\right\rangle = a/\sqrt{1+\sigma^2}\right)\right.$$

where $\begin{bmatrix} \cdot \\ \cdot \end{bmatrix}$ denotes the vector-valued result of concatenating vectors and scalars. For readability, we set $\mathbf{Z}_c = \begin{bmatrix} \mathbf{Z} \\ \epsilon/\sigma \end{bmatrix}$ and $\boldsymbol{\beta}_c = \begin{bmatrix} \boldsymbol{\beta} \\ \sigma \end{bmatrix}/\sqrt{1+\sigma^2}$.

Given that $\boldsymbol{\beta}_c$ is unit-norm (by supposition, $\boldsymbol{\beta}$ is unit-norm), and every element of $\mathbf{Z}_c$ is $\sim \mathcal{N}(0, 1)$ (even $\epsilon/\sigma$), we can easily split a conditional random vector containing $Z_j$ into a conditionally dependent component and independent component:

$$\mathbf{Z}_c\left|\left(\langle \mathbf{Z}_c, \boldsymbol{\beta}_c\rangle = a/\sqrt{1+\sigma^2}\right)\overset{d}{=} (\mathbf{I} - \boldsymbol{\beta}_c\boldsymbol{\beta}_c^\top)\mathbf{Z}'' + \boldsymbol{\beta}_c a/\sqrt{1+\sigma^2}.\right.$$

The first term is orthogonal to $\boldsymbol{\beta}_c$ and so it is the part of $\mathbf{Z}_c$ that is not subject to the condition. In the unconditional case, $\mathbf{Z}_c \sim \mathcal{N}(\mathbf{0}, \mathbf{I})$ and so $\mathbf{Z}'' \sim \mathcal{N}(\mathbf{0}, \mathbf{I})$.

The second term comes from the part of $\mathbf{Z}_c$ that is in the direction of $\boldsymbol{\beta}_c$. Solving for $\mathbf{Z}_c$ where $\langle \mathbf{Z}_c, \boldsymbol{\beta}_c\rangle = a/\sqrt{1+\sigma^2}$ in the case where $\mathbf{Z}_c$ and $\boldsymbol{\beta}_c$ are parallel, we get a constant vector: $\boldsymbol{\beta}_c a/\sqrt{1+\sigma^2}$

Now, we focus on finding $\mathbf{Z}_c|\langle \mathbf{Z}_c, \boldsymbol{\beta}_c\rangle = a/\sqrt{1+\sigma^2}$ for a single index $j$. We have (for $C$ defined to be the dimensionality of $\boldsymbol{\beta}_c$):

$$((\mathbf{I} - \boldsymbol{\beta}_c\boldsymbol{\beta}_c^\top)\mathbf{Z}'')_j + \left(\boldsymbol{\beta}_c a/\sqrt{1+\sigma^2}\right)_j = Z_j''(1 - \beta_{cj}^2) - \sum_{\substack{1 \leq i \leq C \\ i \neq j}} Z_i''\beta_{cj}\beta_{ci} + \beta_{cj}a/\sqrt{1+\sigma^2}$$

Now, note that $Z_j''(1 - \beta_{cj}^2) - \sum_{\substack{1 \leq i \leq C \\ i \neq j}} Z_i''\beta_{cj}\beta_{ci}$ is the sum of independent zero-mean Gaussians, so it itself is a zero-mean Gaussian with variance:

$$(1 - \beta_{cj}^2)^2 + \sum_{\substack{1 \leq i \leq C \\ i \neq j}} \beta_{cj}^2\beta_{ci}^2 = 1 - 2\beta_{cj}^2 + \beta_{cj}^4 + \sum_{\substack{1 \leq i \leq C \\ i \neq j}} \beta_{cj}^2\beta_{ci}^2$$

$$= 1 - 2\beta_{cj}^2 + \beta_{cj}^2\sum_{1 \leq i \leq C} \beta_{ci}^2$$

$$= 1 - \beta_{cj}^2,$$

where we simplified the expression by recalling that $\boldsymbol{\beta}_c$ is unit norm. So we have that $Z_j$ is:

$$Z'\sqrt{1 - \beta_{cj}^2} + \beta_{cj}a/\sqrt{1+\sigma^2} = Z'\sqrt{1 - \frac{\beta_j^2}{1+\sigma^2}} + \frac{\beta_j a}{1+\sigma^2},$$

For $Z' \sim \mathcal{N}(0, 1)$. Now, the problem reduces to finding:

$$\mathbb{E}\left[\Phi\left(Z'\sqrt{1 - \frac{\beta_j^2}{1+\sigma^2}} + \frac{\beta_j a}{1+\sigma^2}\right)\right]$$

By Lemma 2, we have that this is:

$$\Phi\left(\frac{\frac{\beta_j a}{1+\sigma^2}}{\sqrt{1 + 1 - \frac{\beta_j^2}{1+\sigma^2}}}\right) = \Phi\left(\frac{\beta_j a}{(1+\sigma^2)\sqrt{2 - \frac{\beta_j^2}{1+\sigma^2}}}\right)$$

## C.2 LEMMA 5

**Statement of Lemma 5** *Suppose $\Phi$ is the standard Gaussian CDF, $Y \sim \mathcal{N}(0, 1+\sigma^2)$, and $b$ and $a$ are constants. Then we have:*

$$\mathbb{E}[\Phi(aY)\Phi(bY)] = \frac{1}{4} + \frac{1}{2\pi}\tan^{-1}\frac{ab}{2\sqrt{1/(2+2\sigma^2)^2 + a^2/(4+4\sigma^2) + b^2/(4+4\sigma^2)}}$$

*Proof:* By the definition of expected value,

$$\mathbb{E}[\Phi(aY)\Phi(bY)] = \int_{-\infty}^{\infty}\Phi(ay)\Phi(by)f_Y(y)dy$$

$$= \int_{-\infty}^{\infty}\frac{1}{2}\left(1 + \text{erf}\left(\frac{ay}{\sqrt{2}}\right)\right)\frac{1}{2}\left(1 + \text{erf}\left(\frac{by}{\sqrt{2}}\right)\right)\frac{1}{\sqrt{2\pi + 2\pi\sigma^2}}e^{-\frac{y^2}{2+2\sigma^2}}dy$$

Now, let's multiply the terms in the integral:

$$= \frac{1}{4}\int_{-\infty}^{\infty}\frac{1}{\sqrt{2\pi + 2\pi\sigma^2}}e^{-\frac{y^2}{2+2\sigma^2}}dy$$

$$+ \frac{1}{4}\int_{-\infty}^{\infty}\text{erf}\left(\frac{ay}{\sqrt{2}}\right)\frac{1}{\sqrt{2\pi + 2\pi\sigma^2}}e^{-\frac{y^2}{2+2\sigma^2}}dy$$

$$+ \frac{1}{4}\int_{-\infty}^{\infty}\text{erf}\left(\frac{by}{\sqrt{2}}\right)\frac{1}{\sqrt{2\pi + 2\pi\sigma^2}}e^{-\frac{y^2}{2+2\sigma^2}}dy$$

$$+ \frac{1}{4}\int_{-\infty}^{\infty}\text{erf}\left(\frac{ay}{\sqrt{2}}\right)\text{erf}\left(\frac{by}{\sqrt{2}}\right)\frac{1}{\sqrt{2\pi + 2\pi\sigma^2}}e^{-\frac{y^2}{2+2\sigma^2}}dy$$

The first term is an integral over the full domain of a Gaussian PDF, so it is just $\frac{1}{4}$. The second and third terms are integrals over the full domains of odd functions, so they evaluate to zero. Overall, we are left with:

$$= \frac{1}{4} + \frac{1}{4}\int_{-\infty}^{\infty}\text{erf}\left(\frac{by}{\sqrt{2}}\right)\text{erf}\left(\frac{by}{\sqrt{2}}\right)\frac{1}{\sqrt{2\pi + 2\pi\sigma^2}}e^{-\frac{y^2}{2+2\sigma^2}}dy$$

To solve the final integral, we note that the conditions are satisfied in our case to use integral 3 in section 2.7.1 of the integral table from Korotkov & Korotkov (2019):

$$\int_{-\infty}^{\infty}\text{erf}(q_1 z)\text{erf}(q_2 z)e^{-qz^2}dz = \frac{2}{\sqrt{q\pi}}\tan^{-1}\frac{q_1 q_2}{\sqrt{q^2 + qq_1^2 + qq_2^2}}$$

After applying this result, we are left with:

$$\frac{1}{4} + \frac{1}{4\sqrt{2\pi + 2\pi\sigma^2}}\frac{2}{\sqrt{q\pi}}\tan^{-1}\frac{ab}{\sqrt{q^2 + qa^2/2 + qb^2/2}}$$

For $q = \frac{1}{2+2\sigma^2}$. Substituting our variables back in and simplifying, we have:

$$\frac{1}{4} + \frac{1}{2\pi}\tan^{-1}\frac{ab}{2\sqrt{1/(2+2\sigma^2)^2 + a^2/(4+4\sigma^2) + b^2/(4+4\sigma^2)}}$$

### C.3 FULL PROOF

Here, we prove:

$$\mathbb{E}[(\Phi_y(y_1) - \Phi_{\mathbf{x}}(\mathbf{x}_1))^2] = \frac{1}{6} - \frac{1}{\pi} \tan^{-1} \frac{\boldsymbol{\theta}^*}{\sqrt{4(1+\sigma^2) - \boldsymbol{\theta}^{*2}}},$$

where $\Phi_y$ is the empirical CDF of the $y$ values, $\Phi_{\mathbf{x}}$ is the elementwise empirical CDF of the $\mathbf{x}$ values, and $\epsilon$ is zero-mean Gaussian noise $\sim \mathcal{N}(0, \sigma^2)$. We can see that this expected value is monotonic with respect to $\boldsymbol{\theta}^*$, because the numerator inside $\tan^{-1}$ is $\boldsymbol{\theta}^*$, and the $\boldsymbol{\theta}^{*2}$ in the denominator just serves to increase the magnitude of the $\tan^{-1}$ expression. We can also check that the overall expected value is never negative by remembering that the largest value at any index of $\boldsymbol{\theta}^*$ can be at most 1, and so $\frac{1}{\pi} \tan^{-1} \frac{\boldsymbol{\theta}^*}{\sqrt{4(1+\sigma^2) - \boldsymbol{\theta}^{*2}}}$ can never be more than $\frac{1}{6}$.

Now, we can see that proving this theorem also shows

$$\mathbb{E}[(\text{rank}(y_1) - \text{rank}(\mathbf{x}_1))^2]$$

is monotonic with respect to $\boldsymbol{\theta}^*$, where $\text{rank}(y_1)$ is the rank of $y_1$ among the $y$ values, and $\text{rank}(\mathbf{x}_1)$ is the elementwise ranks of $\mathbf{x}_1$ among the $\mathbf{x}$'s. We can then see that Spearman's rank correlation, in expectation, is monotonic with respect to $\boldsymbol{\theta}^*$ (as long as the ranks are distinct), per the following equation for Spearman's rank correlation (Spearman, 1904):

$$r_s = 1 - \frac{6 \sum_{i=1}^{N} (\text{rank}(y_i) - \text{rank}(\mathbf{x}_i))^2}{N(N^2 - 1)}.$$

We begin our proof by finding the value for a single index of $\mathbf{x}_1$, multiplying out the terms, and using linearity of expectation:

$$\mathbb{E}[(\Phi_y(y_1) - \Phi_{x_{1,j}}(x_{1,j}))^2] = \mathbb{E}[\Phi_y(y_1)^2 - 2\Phi_y(y_1)\Phi_{x_{1,j}}(x_{1,j}) + \Phi_{x_{1,j}}(x_{1,j})^2]$$
$$= \mathbb{E}[\Phi_y(y_1)^2] - 2\mathbb{E}[\Phi_y(y_1)\Phi_{x_{1,j}}(x_{1,j})] + \mathbb{E}[\Phi_{x_{1,j}}(x_{1,j})^2].$$

Note that $\mathbb{E}[\Phi_{x_{1,j}}(x_{1,j})^2]$ and $\mathbb{E}[\Phi_y(y_1)^2]$ are both

$$\mathbb{E}[U^2] = \int_0^1 u^2 du = \frac{1}{3},$$

where $U \sim \text{Uniform}(0, 1)$. Now we move to finding $\mathbb{E}[\Phi_y(y_1)\Phi_{x_{1,j}}(x_{1,j})]$. First notice that

$$\mathbb{E}[\Phi_y(y_1)\Phi_{x_{1,j}}(x_{1,j})] = \mathbb{E}[\Phi_{y'}(y')\Phi_{x_{1,j}}(x_{1,j})],$$

where $y' = f^{-1}(y_1)$. This is because $\Phi_y(y_1) = P(Y \leq y_1) = P(f^{-1}(Y) \leq f^{-1}(y_1)) = \Phi_{y'}(y')$. So, we now focus on finding $\mathbb{E}[\Phi_{y'}(y')\Phi_{x_{1,j}}(x_{1,j})]$. By the law of total expectation, it is:

$$\mathbb{E}[\mathbb{E}[\Phi_{y'}(c)\Phi_{x_{1,j}}(x_{1,j})|y' = c]] = \mathbb{E}[\Phi_{y'}(y')\mathbb{E}[\Phi_{x_{1,j}}(x_{1,j})|y' = c]].$$

Note that $y' = \langle \boldsymbol{\theta}^*, \mathbf{x}_1 \rangle + \epsilon$ and $||\boldsymbol{\theta}^*||_2 = 1$, so $y' \sim \mathcal{N}(0, 1 + \sigma^2)$. And $x_{1,j} \sim \mathcal{N}(0, 1)$. So, by Lemma 4, we have that this equals:

$$\mathbb{E}[\Phi_{y'}(y')\Phi(ky')],$$

where $k = \frac{\theta_j^*}{(1+\sigma^2)\sqrt{2 - \frac{\theta_j^{*2}}{1+\sigma^2}}}.$

Also, because $\Phi_{y'}(y') = P(Y' \leq y') = \Phi(\frac{y'}{\sqrt{1+\sigma^2}})$, we can further simplify the expression:

$$\mathbb{E}[\Phi(py')\Phi(ky')],$$

where $p = \frac{1}{\sqrt{1+\sigma^2}}.$

Now, we can write this expected value as an integral and solve it. Via Lemma 5, the solution is:

$$\frac{1}{4} + \frac{1}{2\pi} \tan^{-1} \frac{pk}{2\sqrt{1/(2 + 2\sigma^2)^2 + p^2/(4 + 4\sigma^2) + k^2/(4 + 4\sigma^2)}}$$

Now we have a solution for every component of the expected value that we hope to find. Simplifying, we are left with:

$$\mathbb{E}[(\Phi_y(y_1) - \Phi_{\mathbf{x}}(\mathbf{x}_1))^2] = \frac{1}{6} - \frac{1}{\pi} \tan^{-1} \frac{\theta_j^*}{\sqrt{4(1+\sigma^2) - \theta_j^{*2}}}$$

# D  OPTIMAL PROJECTED WEIGHTS SOLUTIONS

## D.1  LINEAR PROJECTION

**Theorem 2** *Suppose we want to solve:*

$$\hat{\boldsymbol{\theta}}^{\mathrm{proj}} = \underset{\boldsymbol{\theta} \in \mathbb{R}^D}{\arg\min} -\langle \boldsymbol{\theta}, \hat{\boldsymbol{\theta}} \rangle,$$

*subject to:*

$$\sum_{i=1}^{D} \theta_i = 1$$

$$0 \leq \theta_i \leq \tau_i, \forall i \in [1, D],$$

*where $\tau_i > 0$ are fixed values. Then, the solution is:*

$$\hat{\theta}_k^{\mathrm{proj}} = \begin{cases} \tau_k & \text{if } \sum_{j:\, r_j(\hat{\theta}_j) \geq r_k(\hat{\theta}_k)} \tau_j \leq 1 \\ 1 - \sum_{j:\, r_j(\hat{\theta}_j) > r_k(\hat{\theta}_k)} \tau_j & \text{if } \sum_{j:\, r_j(\hat{\theta}_j) \geq r_k(\hat{\theta}_k)} \tau_j \geq 1 \wedge \sum_{j:\, r_j(\hat{\theta}_j) > r_k(\hat{\theta}_k)} \tau_j \leq 1 \\ 0 & \text{otherwise} \end{cases}, \quad (10)$$

*where $r$ breaks ties between $\hat{\theta}_j$ and $\hat{\theta}_k$ for $k \neq j$; otherwise $r$ keeps ordinal relationships the same.*

*Proof:* We proceed by considering each of the three cases from Equation 10.

**Case 1.** Suppose for the sake of contradiction that the optimal solution is $\hat{\boldsymbol{\theta}}^{\mathrm{proj}}$ and yet $\hat{\theta}_k^{\mathrm{proj}} < \tau_k$ for some $\hat{\theta}_k^{\mathrm{proj}}$ falling under the first case of Equation 10. Now suppose that we construct a $\boldsymbol{\theta}'$ also satisfying the projection constraints that is the same as $\hat{\boldsymbol{\theta}}^{\mathrm{proj}}$ except in these places:

$$\theta_k' = \hat{\theta}_k^{\mathrm{proj}} + \Delta = \tau_k$$
$$\theta_p' = \hat{\theta}_p^{\mathrm{proj}} - \delta_1 \geq 0$$
$$\vdots$$
$$\theta_q' = \hat{\theta}_q^{\mathrm{proj}} - \delta_n \geq 0$$

for some $\Delta = \sum_{i=1}^{n} \delta_i > 0$ where $\hat{\theta}_p \geq \cdots \geq \hat{\theta}_q$ are all of the $\hat{\theta}$ values which do not fall under the first condition and where the corresponding $\hat{\theta}^{\mathrm{proj}}$ values are nonzero. We know that there must be some $\hat{\theta}_p^{\mathrm{proj}}, \cdots, \hat{\theta}_q^{\mathrm{proj}}$ from which we can subtract $\delta_1, \cdots, \delta_n$ (and so from which we can take the $\Delta$) because $\sum_{j:\, r_j(\hat{\theta}_j) \geq r_k(\hat{\theta}_k)} \tau_j \leq 1$. Now, we have:

$$\langle \hat{\boldsymbol{\theta}}, \hat{\boldsymbol{\theta}}^{\mathrm{proj}} \rangle - \langle \hat{\boldsymbol{\theta}}, \boldsymbol{\theta}' \rangle$$
$$= \hat{\theta}_k \hat{\theta}_k^{\mathrm{proj}} + \hat{\theta}_p \hat{\theta}_p^{\mathrm{proj}} + \cdots + \hat{\theta}_q \hat{\theta}_q^{\mathrm{proj}} - \hat{\theta}_k \hat{\theta}_k^{\mathrm{proj}} - \hat{\theta}_k \Delta - \hat{\theta}_p \hat{\theta}_p^{\mathrm{proj}} - \cdots - \hat{\theta}_q \hat{\theta}_q^{\mathrm{proj}} + \hat{\theta}_p \delta_1 + \cdots + \hat{\theta}_q \delta_n$$
$$= -\hat{\theta}_k \Delta + \hat{\theta}_p \delta_1 + \cdots + \hat{\theta}_q \delta_n$$
$$\leq \hat{\theta}_p (\delta_1 + \cdots + \delta_n) - \hat{\theta}_k \Delta$$
$$= \hat{\theta}_p \Delta - \hat{\theta}_k \Delta$$
$$\leq 0.$$

At this point, the only way to avoid the contradiction result would be if $\hat{\theta}_k = \hat{\theta}_p = \cdots = \hat{\theta}_q$. Otherwise, the above non-strict inequality would be a strict inequality. If $\hat{\theta}_k = \hat{\theta}_p = \cdots = \hat{\theta}_q$, then we know that $\hat{\theta}_k$ is the smallest $\hat{\theta}$ value satisfying condition 1 and all of the other greater $\hat{\theta}$ values satisfying condition 1 must be projected to their $\tau$ threshold value (otherwise we would get the contradiction result). In this edge case can see above that rearranging the remaining weight among equal $\hat{\theta}$ values does not change the dot product, so all of the solutions that we can get without the contradiction result are equivalently optimal (including the solution from Equation 10).

**Case 3.** This is analogous to case 1. Suppose for the sake of contradiction that the optimal solution is $\hat{\boldsymbol{\theta}}^{\text{proj}}$ and yet $\hat{\theta}_k^{\text{proj}} > 0$ for some $\hat{\theta}_k^{\text{proj}}$ falling under the third case of Equation 10. Now suppose that we construct a $\boldsymbol{\theta}'$ also satisfying the projection constraints that is the same as $\hat{\boldsymbol{\theta}}^{\text{proj}}$ except in these places:

$$\theta_k' = \hat{\theta}_k^{\text{proj}} - \Delta = 0$$
$$\theta_p' = \hat{\theta}_p^{\text{proj}} + \delta_1 \le \tau_p$$
$$\vdots$$
$$\theta_q' = \hat{\theta}_q^{\text{proj}} + \delta_n \le \tau_q$$

for some $\Delta = \sum_{i=1}^n \delta_i > 0$ where $\hat{\theta}_p \ge \cdots \ge \hat{\theta}_q$ are all of the $\hat{\theta}$ values which do not fall under the third condition and where the corresponding $\hat{\theta}^{\text{proj}}$ values are not at their thresholds. By construction we know that there must be some $\hat{\theta}_p^{\text{proj}}, \cdots, \hat{\theta}_q^{\text{proj}}$ to which we can add $\delta_1, \cdots, \delta_n$. Now, we have:

$$
\begin{aligned}
&\langle \hat{\boldsymbol{\theta}}, \hat{\boldsymbol{\theta}}^{\text{proj}} \rangle - \langle \hat{\boldsymbol{\theta}}, \boldsymbol{\theta}' \rangle \\
&= \hat{\theta}_k \hat{\theta}_k^{\text{proj}} + \hat{\theta}_p \hat{\theta}_p^{\text{proj}} + \cdots + \hat{\theta}_q \hat{\theta}_q^{\text{proj}} - \hat{\theta}_k \hat{\theta}_k^{\text{proj}} + \hat{\theta}_k \Delta - \hat{\theta}_p \hat{\theta}_p^{\text{proj}} - \cdots - \hat{\theta}_q \hat{\theta}_q^{\text{proj}} - \hat{\theta}_p \delta_1 - \cdots - \hat{\theta}_q \delta_n \\
&= \hat{\theta}_k \Delta - \hat{\theta}_p \delta_1 - \cdots - \hat{\theta}_q \delta_n \\
&\le -\hat{\theta}_q (\delta_1 + \cdots + \delta_n) + \hat{\theta}_k \Delta \\
&= -\hat{\theta}_q \Delta + \hat{\theta}_k \Delta \\
&\le 0.
\end{aligned}
$$

At this point, the only way to avoid the contradiction result would be if $\hat{\theta}_k = \hat{\theta}_p = \cdots = \hat{\theta}_q$. Otherwise, the above non-strict inequality would be a strict inequality. If $\hat{\theta}_k = \hat{\theta}_p = \cdots = \hat{\theta}_q$, then we know that $\hat{\theta}_k$ is the largest $\hat{\theta}$ value satisfying condition 3 and all of the other smaller $\hat{\theta}$ values satisfying condition 3 must be projected to 0 (otherwise we would get the contradiction result). In this edge case, we can see above that rearranging the remaining weight among equal $\hat{\theta}$ values does not change the dot product, so all of the solutions that we can get without the contradiction result are equivalently optimal (including the solution from Equation 10).

**Case 2.** Above, we show that both Case 1 and Case 3 are true. So, the remaining weight must be given to the single value of $\hat{\boldsymbol{\theta}}^{\text{proj}}$ not covered by either case.

### D.2 QUADRATIC PROJECTION

#### D.2.1 LEMMA 4

**Statement of Lemma 6** *Suppose that $\hat{\boldsymbol{\theta}}^{\text{proj}}$ is the optimal solution to:*

$$\hat{\boldsymbol{\theta}}^{\text{proj}} = \arg\min_{\boldsymbol{\theta} \in \mathbb{R}^D} ||\hat{\boldsymbol{\theta}} - \boldsymbol{\theta}||_2^2,$$

*subject to:*

$$\sum_{i=1}^D \theta_i = 1$$
$$0 \le \theta_i \le \tau_i, \forall i \in [1, D],$$

*where $\tau_i > 0$ are fixed values. Then, $\hat{\theta}_s^{\text{proj}} = 0$ implies that any $j$ with $\hat{\theta}_s > \hat{\theta}_j$ must have $\hat{\theta}_j^{\text{proj}} = 0$.*

*Proof:* This is similar to Lemma 2 from Shalev-Shwartz & Singer (2006). Assume for the sake of contradiction $\hat{\theta}_s^{\text{proj}} = 0$ and $\hat{\theta}_s > \hat{\theta}_j$, yet we have $\hat{\theta}_j^{\text{proj}} > 0$.

Now we can construct another vector $\boldsymbol{\theta}'$ that is the same as $\hat{\boldsymbol{\theta}}^{\text{proj}}$, except in two places:

$$\theta'_s = \hat{\theta}^{\text{proj}}_s + \Delta$$
$$\theta'_j = \hat{\theta}^{\text{proj}}_j - \Delta,$$

for some $\Delta$ satisfying $0 < \Delta < \min(\hat{\theta}^{\text{proj}}_j, \tau_s - \hat{\theta}^{\text{proj}}_s)$. This bound on $\Delta$ ensures that $\boldsymbol{\theta}'$ is still within the thresholds. We know that $\Delta$ can exist because $\min(\hat{\theta}^{\text{proj}}_j, \tau_s - \hat{\theta}^{\text{proj}}_s) > 0$ (by supposition, $\tau_s - \hat{\theta}^{\text{proj}}_s = \tau_s - 0 > 0$ and $\hat{\theta}^{\text{proj}}_j > 0$).

Now we can compute:

$$
\begin{aligned}
||\hat{\boldsymbol{\theta}} - \hat{\boldsymbol{\theta}}^{\text{proj}}||_2^2 - ||\hat{\boldsymbol{\theta}} - \boldsymbol{\theta}'||_2^2 &= (\hat{\theta}_s - \hat{\theta}^{\text{proj}}_s)^2 + (\hat{\theta}_j - \hat{\theta}^{\text{proj}}_j)^2 - (\hat{\theta}_s - (\hat{\theta}^{\text{proj}}_s + \Delta))^2 - (\hat{\theta}_j - (\hat{\theta}^{\text{proj}}_j - \Delta))^2 \\
&= 2\Delta((\hat{\theta}_s - \hat{\theta}^{\text{proj}}_s) - (\hat{\theta}_j - \hat{\theta}^{\text{proj}}_j) - \Delta) \\
&> 2\Delta((\hat{\theta}_s - \hat{\theta}^{\text{proj}}_s) - (\hat{\theta}_j - \hat{\theta}^{\text{proj}}_j) - \min(\hat{\theta}^{\text{proj}}_j, \tau_s - \hat{\theta}^{\text{proj}}_s)) \\
&\geq 2\Delta((\hat{\theta}_s - \hat{\theta}^{\text{proj}}_s) - (\hat{\theta}_j - \hat{\theta}^{\text{proj}}_j) - \hat{\theta}^{\text{proj}}_j) \\
&= 2\Delta(\hat{\theta}_s - \hat{\theta}_j) \\
&> 0.
\end{aligned}
$$

So $\hat{\boldsymbol{\theta}}^{\text{proj}}$ cannot be the optimal solution.

### D.2.2   LEMMA 5

**Statement of Lemma 7** *Suppose that $\hat{\boldsymbol{\theta}}^{\text{proj}}$ is the optimal solution to:*

$$\hat{\boldsymbol{\theta}}^{\text{proj}} = \underset{\boldsymbol{\theta} \in \mathbb{R}^D}{\arg\min} \, ||\hat{\boldsymbol{\theta}} - \boldsymbol{\theta}||_2^2,$$

*subject to:*

$$\sum_{i=1}^{D} \theta_i = 1$$

$$0 \leq \theta_i \leq \tau_i, \forall i \in [1, D],$$

*where $\tau_i > 0$ are fixed values. Then, $\hat{\theta}^{\text{proj}}_s = \tau_s$ implies $\hat{\theta}^{\text{proj}}_j = \tau_j$ for any $\hat{\theta}_j - \tau_j > \hat{\theta}_s - \tau_s$.*

*Proof:* Again, this is similar to Lemma 2 from Shalev-Shwartz & Singer (2006). Assume for the sake of contradiction $\hat{\theta}^{\text{proj}}_s = \tau_s$ and $\hat{\theta}_j - \tau_j > \hat{\theta}_s - \tau_s$, yet we have $\hat{\theta}^{\text{proj}}_j < \tau_j$.

Now we can construct another vector $\boldsymbol{\theta}'$ that is the same as $\hat{\boldsymbol{\theta}}^{\text{proj}}$, except in two places:

$$\theta'_s = \hat{\theta}^{\text{proj}}_s - \Delta$$
$$\theta'_j = \hat{\theta}^{\text{proj}}_j + \Delta,$$

for some $\Delta$ satisfying $0 < \Delta < \min(\hat{\theta}^{\text{proj}}_s, \tau_j - \hat{\theta}^{\text{proj}}_j)$. This bound on $\Delta$ ensures that $\boldsymbol{\theta}'$ is still within the thresholds. We know that $\Delta$ can exist because $\min(\hat{\theta}^{\text{proj}}_s, \tau_j - \hat{\theta}^{\text{proj}}_j) > 0$ (by supposition, $\tau_j - \hat{\theta}^{\text{proj}}_j > 0$ and $\hat{\theta}^{\text{proj}}_s = \tau_s > 0$).

Now we can compute:

$$
\begin{aligned}
||\hat{\boldsymbol{\theta}} - \hat{\boldsymbol{\theta}}^{\text{proj}}||_2^2 - ||\hat{\boldsymbol{\theta}} - \boldsymbol{\theta}'||_2^2 &= (\hat{\theta}_s - \hat{\theta}^{\text{proj}}_s)^2 + (\hat{\theta}_j - \hat{\theta}^{\text{proj}}_j)^2 - (\hat{\theta}_s - (\hat{\theta}^{\text{proj}}_s - \Delta))^2 - (\hat{\theta}_j - (\hat{\theta}^{\text{proj}}_j + \Delta))^2 \\
&= 2\Delta((\hat{\theta}_j - \hat{\theta}^{\text{proj}}_j) - (\hat{\theta}_s - \hat{\theta}^{\text{proj}}_s) - \Delta) \\
&> 2\Delta((\hat{\theta}_j - \hat{\theta}^{\text{proj}}_j) - (\hat{\theta}_s - \hat{\theta}^{\text{proj}}_s) - \min(\hat{\theta}^{\text{proj}}_s, \tau_j - \hat{\theta}^{\text{proj}}_j)) \\
&\geq 2\Delta((\hat{\theta}_j - \hat{\theta}^{\text{proj}}_j) - (\hat{\theta}_s - \hat{\theta}^{\text{proj}}_s) - (\tau_j - \hat{\theta}^{\text{proj}}_j)) \\
&= 2\Delta((\hat{\theta}_j - \tau_j) - (\hat{\theta}_s - \hat{\theta}^{\text{proj}}_s)) \\
&= 2\Delta((\hat{\theta}_j - \tau_j) - (\hat{\theta}_s - \tau_s)) \\
&> 0.
\end{aligned}
$$

So $\hat{\boldsymbol{\theta}}^{\text{proj}}$ cannot be the optimal solution.

### D.2.3 Full Proof

**Theorem 3** *Suppose we want to solve:*

$$\hat{\boldsymbol{\theta}}^{\text{proj}} = \arg\min_{\boldsymbol{\theta}\in\mathbb{R}^D} ||\hat{\boldsymbol{\theta}} - \boldsymbol{\theta}||_2^2,$$

*subject to:*

$$\sum_{i=1}^{D} \theta_i = 1$$

$$0 \leq \theta_i \leq \tau_i, \forall i \in [1, D],$$

*where $\tau_i > 0$ are fixed values. Then the solution is:*

$$\hat{\theta}_k^{\text{proj}} = \min(\max(\hat{\theta}_k - \lambda, 0), \tau_k),$$

*where $\lambda$ is found (through e.g. bisection search) to satisfy:*

$$\sum_{i=1}^{D} \min(\max(\hat{\theta}_i - \lambda, 0), \tau_i) = 1.$$

*Proof:* Note that this problem is the same as the simplex projection problem from Shalev-Shwartz & Singer (2006) and Duchi et al. (2008), except here we have additional $\theta_i \leq \tau_i$ constraints. The Lagrangian for this problem is[4]:

$$\mathcal{L}(\boldsymbol{\theta}, \mu, \zeta, \lambda) = \frac{1}{2}||\hat{\boldsymbol{\theta}} - \boldsymbol{\theta}||_2^2 + \lambda\left(-1 + \sum_{i=1}^{N}\theta_i\right) - \langle\mu, \boldsymbol{\theta}\rangle + \langle\zeta, \boldsymbol{\theta} - \tau\rangle.$$

To find the optimality condition with respect to a single index of $\boldsymbol{\theta}$, we set the derivative to zero:

$$\frac{d\mathcal{L}}{d\theta_i} = \theta_i - \hat{\theta}_i + \lambda - \mu_i + \zeta_i = 0.$$

The complimentary slackness KKT condition gives us that $\zeta_i = \mu_i = 0$ when $0 < \theta_i < \tau_i$, so for $\theta_i$ not at the boundary of our constraints, we get:

$$\theta_i = \hat{\theta}_i - \lambda.$$

So, we have that for all $\theta_i \in (0, \tau_i)$, there is a shared value $\lambda$ which we subtract from $\hat{\theta}_i$ to get the value of $\theta_i$. How do we know which $\theta_i$ are 0 and which $\theta_i$ are $\tau_i$, though?

Assume that we know $\lambda$. By Lemma 6, we can characterize the optimal solution as:

$$\hat{\theta}_k^{\text{proj}} = \max(\hat{\theta}_k - \lambda, 0),$$

for $\hat{\theta}_k^{\text{proj}} \neq \tau_k$. By Lemma 7, we can characterize the optimal solution as:

$$\hat{\theta}_k^{\text{proj}} = \min(\hat{\theta}_k - \lambda, \tau_k),$$

for $\hat{\theta}_k^{\text{proj}} \neq 0$. So, we can combine these two forms to get:

$$\hat{\theta}_k^{\text{proj}} = \min(\max(\hat{\theta}_k - \lambda, 0), \tau_k).$$

Now recall that we have the following constraint:

$$\sum_{i=1}^{D} \min(\max(\hat{\theta}_i - \lambda, 0), \tau_i) = 1.$$

Given this constraint, we can find $\lambda$ through search (moving the value up or down). We can see this by noticing that $\sum_{i=1}^{D}\min(\max(\hat{\theta}_i - \lambda, 0), \tau_i)$ is a strictly decreasing function of $\lambda$ between the setting of $\lambda$ that makes $\hat{\theta}_i - \lambda > 0$ for at least one $i$, and the setting of $\lambda$ that makes $\hat{\theta}_i - \lambda < \tau_i$ for at least one $i$. So in this range, there is only one setting of $\lambda$ that satisfies this equation. We can only choose a $\lambda$ outside of this range when $\sum_{i=1}^{D}\tau_i = 1$, and in this case the solution is trivial: $\hat{\theta}_i^{\text{proj}} = \tau_i$ for all $i$.

---

[4]Note that multiplying $||\hat{\boldsymbol{\theta}}^{\text{proj}} - \boldsymbol{\theta}||_2^2$ by $\frac{1}{2}$ does not change the minimization problem and enables us to get rid of a factor of 2 after taking the derivative of the Lagrangian.

# E    ALTERNATIVE METHODS

Our estimator is far from the only reasonable high-dimensional, single-index model estimator. We briefly discuss some alternatives and the tradeoffs involved before moving to experimental results.

We could use classic low-dimensional methods regularized for the high-dimensional setting. This includes ordinal regression (Wooldridge, 2010) and the isotron algorithm (Kalai & Sastry, 2009). We found these methods to underperform correlation-based estimators, and tuning hyperparameters added additional complexity that was not needed in the correlation-based approaches.

Another class of methods involve scaling laws (Kaplan et al., 2020; Llama Team, 2024; Ruan et al., 2024). We could transform the $y$ values via an inverse sigmoid or power law, and fit high-dimensional linear regression methods (e.g. ridge, partial least squares, or Lasso). We initially found this approach promising, but the inverse transforms were unstable, and the combination of fitting the nonlinear transform and regularization required significant amounts of tuning.

Rank-correlation methods, including our robustified version of the estimator from Chen & Banerjee (2017), and even the standard Spearman correlation (Spearman, 1904) (see Appendix H) performed well. We believe that in general, robust per-feature correlations are likely to perform well as $D \gg N$, and extreme levels of regularization are needed to obtain reasonable models. Sparse methods such as the Lasso (Tibshirani, 1996) are one classic answer, but we cannot necessarily assume that the underlying correlations $\theta^*$ are sparse, and we did not find these techniques to perform well.

# F    LOSS MATRIX COMPUTATION SPECIFICS

For all of our initial experiments, we computed the loss matrix as follows. For efficiency purposes, we sampled only 25 pages for a domain's bits-per-byte (BPB) computation even if a domain had more than 25 pages. To get an LLM's BPB on a page, we split the page into chunks of text that were 512 tokens according to a reference tokenizer (we used the Llama 2 7B tokenizer; Touvron et al. 2023). These text chunks turned out to be small enough to fit in the context of every LLM we tested. We then averaged BPB across chunks for each page and then across pages for each domain.

# G    ADDITIONAL DETAILS FOR INITIAL PRETRAINING EXPERIMENTS

In this section, we specify hyperparameters and methods used for LLM pretraining and evaluation for our initial LLM pretraining experiments. We also specify settings used for the data-selection methods.

## G.1    LLM PRETRAINING

We trained each LLM on 4 NVIDIA A100 GPUs. At 3.2B tokens, each training run took under 3 hours with the Hugging Face Trainer (Wolf et al., 2019) and appropriate PyTorch (Ansel et al., 2024) compile flags. We provide pretraining hyperparameters in Table 2. Given our per-device batch size, we found the learning rate by increasing it by a factor of 2 until we saw instability and then using the highest learning rate where no instability was observed. Refer to the Pythia paper (Biderman et al., 2023) for more information; we initialized the model from scratch using their 160M model configuration at `https://huggingface.co/EleutherAI/pythia-160m`. Other hyperparameters can be assumed to be Hugging Face Trainer defaults at the time of this writing.

## G.2    LLM EVALUATION

At the end of the pretraining script, we used the Eleuther AI Eval Harness (Gao et al., 2023). For efficiency, we set the sample limit to 5000 examples per benchmark. Elsewhere, we used the default settings. On 4 NVIDIA A100s, it took only a few minutes per LLM to compute evaluation results for SciQ, ARC Easy, PIQA, LAMBADA, and all of the translations of LAMBADA.

## G.3    DSIR

DSIR (Xie et al., 2023b), despite its simplicity, requires some tuning. A decision must be made about how to format the bemchmark data into a single piece of text per example so that it can be

Table 2: LLM Pretraining Hyperparameters

| Parameter | Value |
|---|---|
| Per-device Batch Size | 128 |
| Learning Rate | $5 \times 10^{-3}$ |
| Warmup Ratio | 0.1 |
| Adam $\beta_1$ | 0.9 |
| Adam $\beta_2$ | 0.95 |
| Adam $\epsilon$ | $1 \times 10^{-8}$ |
| Weight Decay | 0.1 |
| LR Scheduler | cosine |
| Max Grad Norm | 1.0 |
| BF 16 | True |
| Distributed Backend | nccl |
| Gradient Accumulation Steps | 1 |

Table 3: Unique pretraining tokens selected per benchmark, from DSIR.

| Benchmark | Tokens |
|---|---|
| ARC Easy | 2,905,206,499 |
| PIQA | 2,910,486,295 |
| SCIQ | 2,920,734,042 |
| LAMBADA | 3,022,219,424 |
| LAMBADA$_{DE}$ | 3,210,986,137 |
| LAMBADA$_{ES}$ | 3,396,528,704 |
| LAMBADA$_{FR}$ | 3,413,930,081 |
| LAMBADA$_{IT}$ | 3,384,854,845 |

compared with potential pretraining data in terms of n-gram overlap. The LAMBADA tasks only have one text column per example, so the decision here is trivial. Examples from the other tasks each have a question, possibly a context, and a set of multiple choice answers to choose from. We chose to concatenate all of these columns together with spaces to form one piece of text per example, duplicating the same question as a prefix for each different answer.

DSIR does not allow the user to specify the exact number of unique tokens desired for pretraining. It only allows the specification of the number of unique pages, which can have wildly varying token counts. For every DSIR job, we set the desired number of pages to 3325589, which we found through binary search to produce slightly more than 3.2B unique tokens for LAMBADA$_{FR}$. It was expensive to find this number for even one bechmark, because for each iteration of the binary search, we had to run DSIR and then the Pythia tokenizer to know how many tokens resulted from the input page number parameter. We provide the number of unique tokens from DSIR for each task in Table 3. We pretrained on 3.2B tokens for every LLM regardless of whether all of them were unique.

### G.4 FASTTEXT

The "SOTA" fastText model from Li et al. (2024) is available here: https://huggingface.co/mlfoundations/fasttext-oh-eli5. We used this model to filter data by sorting pages by the

model's "high quality" score, including the top pages in order until we had either reached or gone slightly over 3.2B unique tokens. This aligns with the data-selection procedure in the original paper, and is also essentially the same as running the linear projection (Equation 10) at the page-level. We also applied this method when selecting data using our own fastText filter trained by our algorithm.

## H  ADDITIONAL INITIAL PRETRAINING RESULTS

In Figure 5, we present additional initial pretraining results for methods in our loss-performance correlation data selection paradigm. We find that using Spearman rank correlation (Spearman, 1904) in place of our estimator achieves comparable performance. On some tests, it performs even better than our estimator. We also find that using the quadratic projection, while perhaps more intuitive, leads to worse performance than the linear projection.

## I  PERFORMANCE PREDICTION SCORES

We include 5-fold leave-out R2 scores for all initial experiment tasks in Figure 6. However, we complement these strong results with the additional observation that simply taking the *mean* loss across all domains is a strong predictor of model performance (bottom row). The surprising effectiveness of average loss over uniformly sampled documents has been discussed extensively (Owen, 2024; Wei et al., 2022; Kaplan et al., 2020) and our results further suggest that regressions with correlations only slightly above the mean loss baseline can still result in effective data selection methods.

## J  PRETRAINING LANGUAGE DISTRIBUTION

Figure 7 shows the language distributions of the projected estimates for our initial RPJv2 experiments. Our algorithm provides significant enrichment of the corresponding languages for the multilingual benchmarks (LAMBADA_*), but it does not *exclusively* select domains in one language. In contrast, for English benchmarks our approach selects nearly exclusively English data, likely due to the large quantity of high-quality English data in our pretraining data pool. There are significantly fewer tokens in non-English languages in the data pool and our $\tau$ constraint prevents their duplication.

Figure 8 shows what the projected estimate in our pretraining experiments would be if we had a pretraining data pool $5\times$ as large. We see here that the estimate does an even better job at selecting pretraining data with the language that matches the target task.

## K  PARAMETER COUNT DISTRIBUTION FOR ESTIMATOR LLMS

In Figure 9, we present the parameter-count histogram of the 90 models from the Open LLM Leaderboard (Beeching et al., 2023) that we used to compute our estimate for pretraining data selection. Only 8 models here are less than 160M parameters. Despite this, our estimate can be used to effectively pretrain 160M parameter LLMs.

## L  ANALYSIS OF THE MODEL-LOSS MATRIX $\mathbf{X}$

What information is contained in the matrix of model losses $\mathbf{X}$? Clearly, it must contain semantically meaningful information about the data, such as the language that a piece of text is in. We performed PCA (Pearson, 1901) and t-SNE (van der Maaten & Hinton, 2008) on $\mathbf{X}$ and plotted the first two components for each of our 9,841 RPJv2 domains. As shown in the first row of Figure 10, we found two components with relatively high singular values. The first component clearly corresponds with the language of a domain. The second component corresponds with the average bits-per-byte or entropy of a domain. The t-SNE components show the same general pattern as well as showing that the language clusters are very well separated. As shown in our plots, there are several salient clusters within the language clusters. Within the English cluster, we found a subcluster for luxury goods, another for legal services and information, another for academic research, and even a cluster for funeral homes.

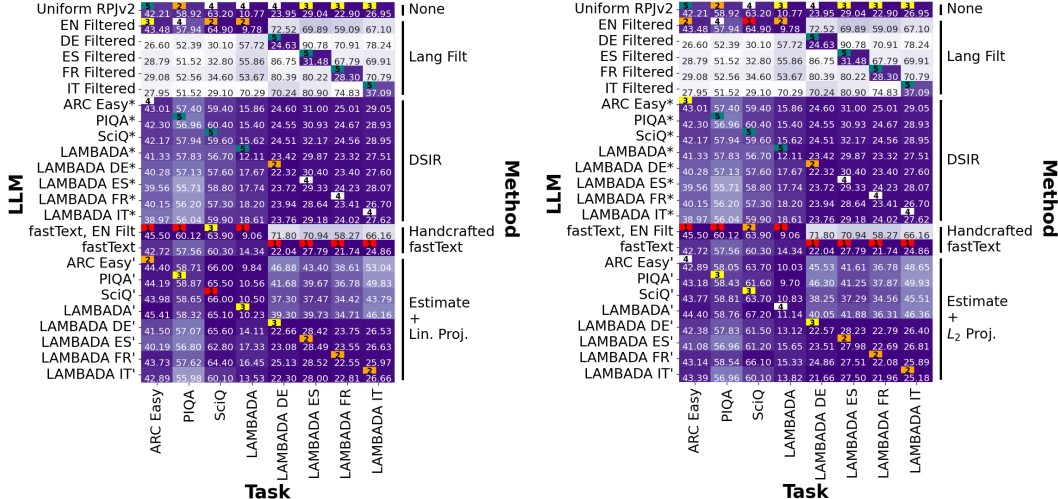

(a) Estimate with linear projection. This is our algorithm from the main text without training the additional fastText filter.

(b) Estimate with quadratic projection. Same as (a) except the linear projection is replaced with the quadratic projection.

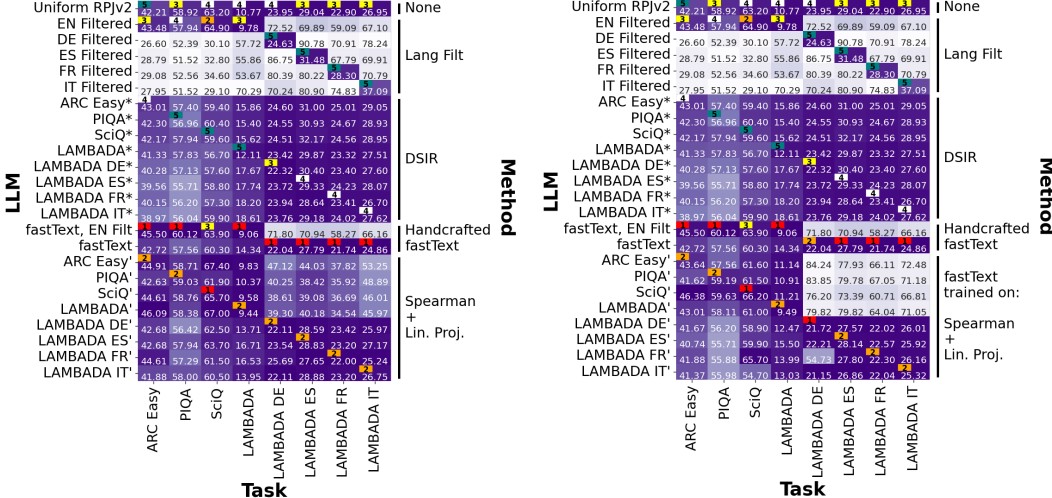

(c) Spearman rank correlation with linear projection. Same as (a) except we replaced our estimator with the Spearman rank correlation.

(d) fastText filter trained on data selected in (c). This is the same as our algorithm in the main text, replacing our estimator with the Spearman rank correlation.

Figure 5: Pretraining results for different methods within our paradigm. Overall, we see that many rank-correlation pretraining data selection approaches perform well.

The second row of Figure 10 shows plots for the loss matrix when we take the principal components of the other dimension, where points correspond to the 90 LLMs. For PCA, PC1 corresponds to entropy. For both cases, it is less clear what the other PCs are, but when we color the three largest families of models in our data (Pythia (Biderman et al., 2023), Qwen (Bai et al., 2023), and OpenLlama (Geng & Liu, 2023)), we see that model families are clustered together in the PC graphs.

## M   SCALED UP EXPERIMENTS

During the ICLR review process, and before we had completed the preregistration in Appendix N, concerns about scale were raised. So, we re-ran the uniform sampling baseline and our main perplexity correlations method to create Figure 11, analogous to Figure 2. Besides scale, the only other

| | ARC Easy | PIQA | SciQ | LAMBADA | LAMBADA DE | LAMBADA ES | LAMBADA FR | LAMBADA IT |
|---|---|---|---|---|---|---|---|---|
| Proj. Estimate | 89.1±3.6 | 89.7±3.0 | 78.1±5.0 | 92.6±1.7 | 89.9±2.4 | 84.3±4.1 | 92.8±1.8 | 91.8±2.0 |
| Estimate | 85.7±4.6 | 86.3±3.8 | 75.2±6.4 | 91.1±2.0 | 89.0±2.3 | 80.6±4.7 | 91.2±2.1 | 92.0±1.8 |
| Mean Loss | 86.1±4.5 | 86.3±3.9 | 75.0±6.3 | 91.3±2.0 | 88.8±2.4 | 80.7±4.8 | 91.6±2.0 | 92.1±1.7 |

Figure 6: Held-out $R^2$ score of our raw correlation estimate $\hat{\theta}$, our projected estimate $\hat{\theta}^{\mathrm{proj}}$, and the average loss baseline. The $95\%$ bootstrapped confidence intervals are wide enough that no individual comparison is significant. Across benchmarks, $\hat{\theta}^{\mathrm{proj}}$ has statistically significant gains over the baseline (p=0.035) as it is unlikely that $\hat{\theta}^{\mathrm{proj}}$ beats mean loss 7 times out of 8 by chance.

| | Uniform RPJv2 | ARC Easy | PIQA | SciQ | LAMBADA | LAMBADA DE | LAMBADA ES | LAMBADA FR | LAMBADA IT |
|---|---|---|---|---|---|---|---|---|---|
| EN | 63.17 | 99.01 | 99.02 | 98.88 | 98.59 | 59.16 | 50.48 | 57.85 | 58.36 |
| DE | 10.54 | 0.42 | 0.46 | 0.67 | 0.60 | 15.01 | 11.97 | 5.26 | 14.98 |
| ES | 9.85 | 0.25 | 0.25 | 0.36 | 0.42 | 7.83 | 14.17 | 14.34 | 8.94 |
| FR | 10.41 | 0.32 | 0.27 | 0.09 | 0.39 | 11.46 | 16.89 | 16.89 | 11.25 |
| IT | 6.03 | 0.00 | 0.00 | 0.00 | 0.00 | 6.53 | 6.48 | 5.65 | 6.48 |

Figure 7: Language distributions of pretraining data selected by perplexity correlations. The default RPJv2 distribution is given in the left column for reference. The English benchmark targets often exclusively select English but the reverse is not the case. In every case, our approach selects more data than the default from the benchmark-matched language (shown as a green box in each column).

feature we changed was the number of pretraining tokens, which we set to 8.2B to keep the ratio chinchilla-optimal (Hoffmann et al., 2022). This setting is arguably harder for perplexity correlations, because the uniform sampling method was allowed to sample from a pool of 8.2B unique tokens, whereas for our perplexity correlations experiment, we duplicated the original 3.2B set of tokens from the 160M parameter model experiments. Despite training on data that is far more duplicated, perplexity correlations still achieves superior performance in 7 out of 8 tasks as seen in Figure 11.

# N SCALING UP FURTHER: PREREGISTERED EXPERIMENTS

In small-scale experiments, our approach is competitive with the leading approach from Li et al.'s survey: a fixed fastText model (Joulin et al., 2016), manually augmented with the best language filtering. This leading approach is heuristic and hand-crafted, requiring appropriate language filtering matched to the target benchmark and assumptions about what good pretraining data looks like. Our approach does not make these assumptions and could potentially improve as more public models are released and we have better data to estimate $\theta^*$.

While our initial results are generally positive, many past data selection methods have reported initially positive results, only to later break: they may fail to scale to larger models or rely on specific details of their experimental setting. Our 160M-scale experiments may also raise such concerns.

We designed a pre-registered scaling experiment that addresses both the concerns of scale and external validity. We used the permanence of arXiv preprints as a mechanism to preregister a series of scaling experiments within the DataComp-LM framework (Li et al., 2024), which is a testbed for data-selection techniques released with the recent survey. Pre-registering held-out scaling experiments committed us to reporting potentially negative results, and avoid overfitting to our chosen experimental settings. Our first arXiv submission with the preregistration is available at https://arxiv.org/abs/2409.05816v1.

DataComp-LM was ideal for this preregistered scaling experiment, as it standardizes the setting by providing a pool of 240 trillion tokens, pretraining code for 412M to 7B parameter models, and evaluation code for 53 benchmarks, 22 of which are labelled as "core" benchmarks that scale predictably. Importantly, we did not train *any* models on DataComp-LM using our methods beforehand or baselines, making this a true held-out experiment with known high-performance baselines.

| | Uniform RPJv2 | ARC Easy | PIQA | SciQ | LAMBADA | LAMBADA DE | LAMBADA ES | LAMBADA FR | LAMBADA IT |
|------|------|------|------|------|------|------|------|------|------|
| EN | 63.17 | 98.39 | 100.00 | 99.94 | 96.69 | 1.09 | 7.64 | 17.27 | 26.35 |
| DE | 10.54 | 1.59 | 0.00 | 0.02 | 0.40 | 67.53 | 2.27 | 2.30 | 27.94 |
| ES | 9.85 | 0.02 | 0.00 | 0.00 | 1.60 | 0.79 | 36.96 | 33.40 | 4.08 |
| FR | 10.41 | 0.00 | 0.00 | 0.04 | 1.31 | 0.66 | 53.07 | 46.97 | 17.49 |
| IT | 6.03 | 0.00 | 0.00 | 0.00 | 0.00 | 29.94 | 0.06 | 0.05 | 24.15 |

Figure 8: This figure is analogous to Figure 7, except the $\tau$ thresholds have been multiplied by 5. We see that our approach selects even more relevant data when the selection pool is larger.

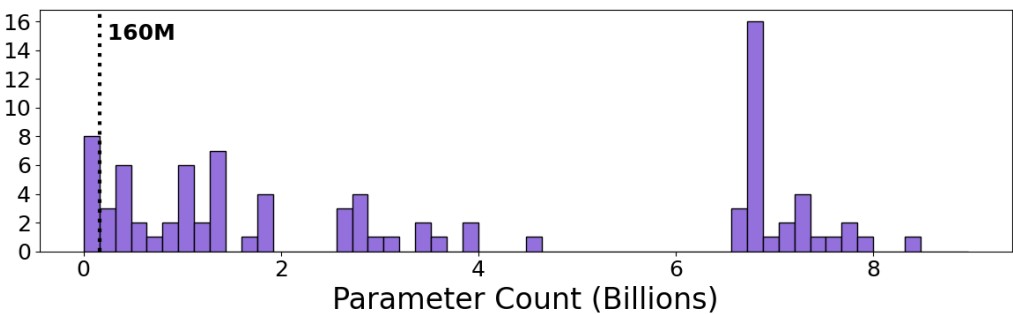

Figure 9: The parameter-count histogram of the 90 models from the Open LLM Leaderboard (Beeching et al., 2023) that we used to compute our estimate for pretraining data selection. Bar widths are 160M. The smallest model in the sample has $\approx$33M parameters and the largest has $\approx$9B. The spike around 6.7B parameters is due to a large number of partially trained Pythia (Biderman et al., 2023) checkpoints from the same training run at that scale. Our algorithm has the hard task of selecting pretraining data for 160M parameter models, which is abnormally small in the set of models used to compute the estimate.

We ran the best-performing approach from our paper: a fastText filter trained on our correlation estimator. We defined the target benchmark for our estimator as the average of the "core" DataComp-LM benchmarks and ran our estimator with perplexities from our set of 90 OpenLM Leaderboard LLMs on a uniform subsample of the DataComp-LM pool of data. Instead of using the provided DCLM code (as planned in our preregistration), we used our own pretraining and eval code after consulting with the DCLM authors, as their codebase could not be easily run outside their infra setup, and they advised us to switch to our own codebase.

Specifically, we used our own code for training Pythia-architecture LLMs (and the same methodology for tuning hyperparameters as in the 160M experiments before), as well as the Eleuther Eval Harness (Gao et al., 2023) for replicating the "core" benchmark evaluations using the settings reported in the DCLM paper. Differences in Pythia architectures and Eleuther Eval Harness implementations unfortunately make our evaluation numbers not directly comparable to the original DCLM codebase numbers.

We report results for the "Filtering 1B-1x" track, where a 1.4B parameter LLM is trained on 28.8B tokens chosen from a 1.64T sample of the DataComp-LM pool of data.[5] In the DataComp-LM paper, they apply their fixed fastText filter as a final step after several complicated deduplication and filtering steps. We report results where our fastText classifier is used as a direct substitute for this last step alone (filtering from the "pre-filtered pool"), as well as another test in which we replace the entire pipeline with one classifier (filtering from the "raw pool").

We also report results where our estimator is trained at the domain-level (following this paper) and where our estimator is trained at the page-level (which we had not tried before).

---

[5]Due to disk constraints, we filtered starting from a 10% random sample of this 1.64T sample for every method.

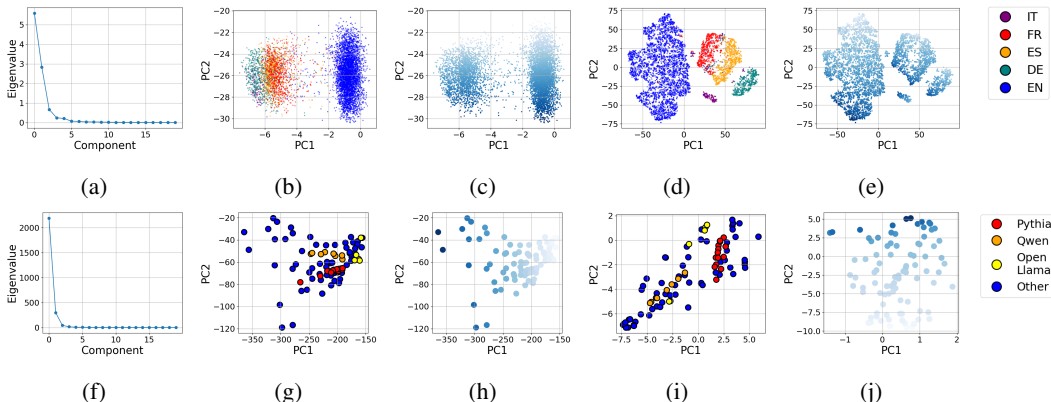

Figure 10: Analysis of the loss matrix. The first row treats domains as examples to be projected via PCA, while the second row treats models as examples. Panels (a): eigenvalue decay for the eigende-composition of the $D \times D$ covariance matrix resulting from the loss matrix; a few dominant PCs are seen. (b) and (c): domains plotted by the first two PCA components showing separation of language in b and entropy in c. (d,e) show analogous plots in t-SNE with a clearer separation of language. (f): eigenvalue decay analogous to (a). (g,h): models plotted by the first two PCA components showing clustering by model family (clusters show Pythia (Biderman et al., 2023), Qwen (Bai et al., 2023), and OpenLlama (Geng & Liu, 2023) derivatives – the three largest clusters in our data), and average model loss. (i,j) show analogous results under t-SNE where (i) is normalized to remove per-model entropy differences.

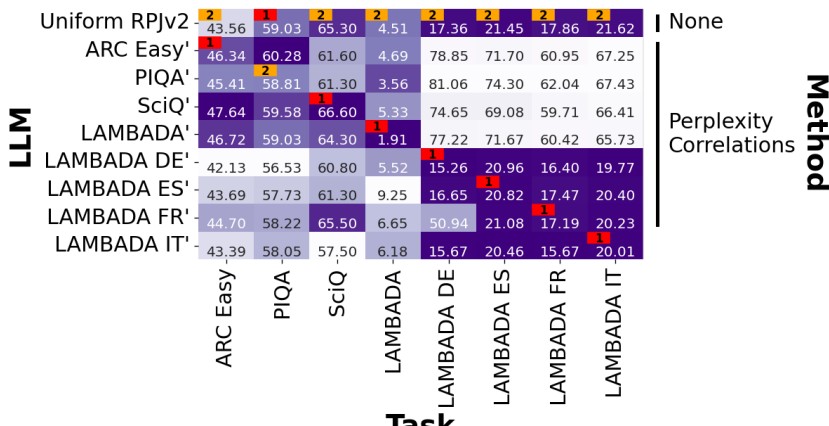

Figure 11: Analog to Figure 2 at the 410M parameter and 8.2B token scale. For perplexity correlations, we duplicate the original 3.2B tokens, whereas the uniform sampling baseline is allowed to sample from 8.2B unique tokens.

Finally, we report analogous results where we replace the "core" evaluation score with the average score on all of the non-English LAMBADA translations, and compare the raw fastText classifier from Li et al. (2024) to our approach, using both of these approaches in place of the full filtering pipeline from 1.64T tokens. We preregistered this additional multilingual task because "core" does not include multilingual evaluations.

Beyond our preregistered experiments, we also ran chinchilla-optimal 160M, 410M, and 1B scales to complement our results. We applied the same perplexity correlations fastText filter at all data selection scales. To train it, we set the token threshold in our algorithm to be 50% of the tokens used for training the estimator, meaning that approximately 50% of the domains/pages from our estimate were labeled as 'include' and the other approximate 50% were labeled as 'exclude' for the purposes of fastText training. To train our estimator, we used approximately the same scale of data as our

initial RPJv2 BPB matrix. For the pre-filtered DCLM pool, this was a sample with about the same disk size as the RPJv2 estimate sample with 129,376 pages and 9,952 domains at $\geq 13$ pages per domain (we found that the pages in this pool were about twice as long on average as the RPJv2 pages). For the raw DCLM pool, this was a sample also of about the same disk size with 325,682 pages and 13,732 domains at $\geq 23$ pages per domain (page lengths were generally shorter in this sample).

All results can be found in Figure 3.

## O  TOP CORRELATED DOMAINS PER-TASK

Here, we list the top 10 most correlated domains (before projection) found by our main rank correlation estimator for the pretraining experiments.

**ARC Easy**. api-bridge.azurewebsites.net, superlogical.net, www.aaeoptometry.com, www.akronchildrens.org, www.edusofttech.com, www.fredericksburgtso.com, www.harborfronthospitalforanimals.com, www.hopkinsallchildrens.org, www.metropolitanvisionnyc.com, www.myheartliveshere.com

**PIQA**. api-bridge.azurewebsites.net, familyserviceshub.havering.gov.uk, ricardofrancia.com, www.aaeoptometry.com, www.akronchildrens.org, www.eczemainfoclub.com, www.groupeproxim.ca, www.gynecology-doctors.com, www.medicineshoppe.ca, www.metropolitanvisionnyc.com

**SciQ**. api-bridge.azurewebsites.net, goodbusinesskit.com, original-ideas.net, pos-university.com, taraweddings.ca, thefloristic.com, www.cuemath.com, www.edusofttech.com, www.groupeproxim.ca, www.landkreis-waldeck-frankenberg.de

**LAMBADA**. 2gringos.blogspot.com, birdingmarc.blogspot.com, books.google.ae, chestofbooks.com, joint-research-centre.ec.europa.eu, snoqualmie.cementhorizon.com, twowheeledmadwoman.blogspot.com, www.ganssle.com, www.sarahhague.com, www.themodernnovel.org

**LAMBADA DE**. 1nselpresse.blogspot.com, biomedicalhouse.com, schariagegner.wordpress.com, truthfriends.us, www.aerzteblatt.de, www.buddha-blog.online, www.deutschesgesundheitsportal.de, www.global2015.net, www.juedische-allgemeine.de, www.metropolis-verlag.de

**LAMBADA ES**. archi7.net, catolico.org, cineclubdecaen.com, dbe.rah.es, www.ca-se-passe-la-haut.fr, www.corsarios.net, www.e-stoire.net, www.la-historiadora.com, www.proverbes-francais.fr, www.vedaveda.com

**LAMBADA FR**. archi7.net, cineclubdecaen.com, es.m.wikipedia.org, images.cnrs.fr, irb-cisr.gc.ca, www.ca-se-passe-la-haut.fr, www.corsarios.net, www.futura-sciences.com, www.neurologia-castellon.es, www.vedaveda.com

**LAMBADA IT**. doc.studenti.it, message-for-you.net, shop.fedecultura.com, www.getstoryshots.com, www.global2015.net, www.peterlang.com, www.scrutatio.it, www.scuolafilosofica.com, www.scuolissima.com, www.storieparallele.it

**Non-EN LAMBADA (DCLM Raw Pool)**. digi.ub.uni-heidelberg.de, it.thefreedictionary.com, it.wikiquote.org, slideplayer.it, www.astro.com, www.epo.org, www.kunsthaus.ch, www.logitech.com, www.peterlang.com, www.zenit.org

**DCLM Core (DCLM Raw Pool)**. au.finance.yahoo.com, eldaddp.azurewebsites.net, nrich.maths.org, serc.carleton.edu, whois.epik.com, www.bom.gov.au, www.countrycurrencyrates.com, www.ecdc.europa.eu, www.iaaf.org, www.metoffice.gov.uk

