# OpenReview forum: "Improving Pretraining Data Using Perplexity Correlations"
_ICLR.cc/2025/Conference — ICLR 2025 Poster_

### Official Review · Reviewer_TyHg · 2024-11-02

**Soundness:** 2
**Presentation:** 2
**Contribution:** 2
**Rating:** 6
**Confidence:** 3

**Summary:**

This paper proposes a novel approach to data selection for pretraining large language models (LLMs) using perplexity correlations. This work is based on a simple observation: LLM losses on many pretraining texts are correlated with downstream benchmark
performance, and selecting high-correlation documents is an effective pertaining data selection method. This approach is compute efficient and has competitive performance.

**Strengths:**

1. Using perplexity correlations for data selection is well-motivated and easy to implement.

2. The paper provides a solid theoretical foundation for the proposed method, including a detailed discussion of the single-index model and the estimation of correlation coefficients. The mathematical derivations and proofs are rigorous and well-presented.

3. In experiments, authors demonstrate that their method outperforms existing data selection techniques in terms of downstream benchmark performance.

**Weaknesses:**

1. While the paper demonstrates the effectiveness of the approach at the 160M parameter scale (though the data selection parameters are generated with LLM of 7B),  how the method would scale to larger models' pertaining remains unclear. The extrapolation from smaller models to larger ones could be a potential limitation.

**Questions:**

N/A

---

> ### Author Response · Authors · 2024-11-22
>
> We appreciate your useful comments about scaling up our experiments, and about using our approach to extrapolate from smaller to larger model scales. See the response to all reviewers where we try to address your concerns.

---

### Official Review · Reviewer_9vdE · 2024-11-03

**Soundness:** 3
**Presentation:** 2
**Contribution:** 3
**Rating:** 5
**Confidence:** 3

**Summary:**

This paper proposes a new framework for pre-training data selection by leveraging large amounts of open-weight language models.  Concretely, a data domain is selected for pre-training when open-weight LLMs’ log probabilities on this domain highly correlate with performance on a target downstream task. The higher correlation a data domain demonstrates, the more weight the domain occupies the pre-training data. The proposed data selection algorithm is evaluated with small models of 160M parameters trained on a data scale of 3.2B tokens. Experimental results show that the proposed data selection algorithm, without relying on hand-crafted heuristics and expensive model training runs, approximately matches the handcrafted fastText baseline.

**Strengths:**

- This paper proposes a new framework for pre-training data selection without extensive manual labor by leveraging large amounts of pre-trained open-weight models.
- The results validated on small 160M parameter models look promising.
- The studied topic is important and relevant to many language model practitioners.

**Weaknesses:**

- It is unclear how well the proposed framework works for larger scale pre-training where multiple static benchmarks should be considered, and when the model size can be larger than the maximum model size (~8B) used for computing perplexity-benchmark correlation.
- The theory section is a bit difficult to follow. In the paragraph above section 5.2 "*This suggests that the performance of our approach is not dependent on the precise form of the estimator that is coupled to our theory results, but holds broadly across perplexity-correlation relationships.*" If the theoretical result is less universal than empirical observation, it can be interesting to add more discussion on the more general relationship.
- The proposed model provides the ordering of pre-training domains, however, from what I understand, the sampling ratio across domains is not determined by the benchmark-perplexity correlation, but rather “*select domains in greatest to least γ, taking all the tokens in each domain once, until we exhaust our total pretraining token budget.*” The discussion about the exact ratio to mix various domains is a little limited.

**Questions:**

- How can the proposed method be extended to selecting data for multiple downstream tasks, including unseen and/or dynamic benchmarks?
- Will the data domains selected using models of a certain scale (e.g., < 8B) be helpful for training larger models (e.g., 70B)?
- How would varying the token ratio of pre-training domains affect the downstream performance? That is, mixing domains beyond "selecting all the tokens in each domain once" until token budget is exhausted.

---

> ### Author Response · Authors · 2024-11-22
>
> These are great points about extrapolation from smaller to larger models, the theory limitations, and the behavior of our approach when aggregating across benchmarks. We have made an attempt to address all of them in our response to all reviewers above. Regarding your comment about experiments that go beyond including or excluding all tokens per-domain: this is definitely a limitation of our approach and we are open to suggestions about how to do this. Unfortunately, we had to make our estimators so robust that we could only trust them up to the ranks. We already attempted to train models with the assumption that we could blindly trust the values instead of just the ranks, by running an L2 projection. We provided the results in the appendix, but unfortunately they show worse performance. It would be fantastic if future work could find an estimator that is very robust and returns trustable values too.

---

> > ### Author Response · Authors · 2024-12-03
> > **Thoughts on our new experimental results?**
> >
> > Thank you again for the feedback - we ran experiments at ~ 2-3x larger scales, and the results remain positive. We are wondering if you have a chance to take a look and have any responses?

---

### Official Review · Reviewer_2JKP · 2024-11-04

**Soundness:** 4
**Presentation:** 3
**Contribution:** 3
**Rating:** 8
**Confidence:** 4

**Summary:**

The authors build on a large number of existing open LMs and the observation that perplexity of an LM on a sample of text is predictive of performance on some (presumably related) benchmarks to produce a method for selecting pretraining data, aiming to optmize downstream performance.  The method doesn’t require any training of models. Specifically they greedily sample all available data, without repetition, from each domain in order of correlation until a desired token budget is met. They also filter at the  webpage level using a FastText classifier trained to discriminate the selected and non-selected domains from the previous step.
They apply this set up over:
- 90 models from 33M to 9B parameters from the Open LLM leaderboard, including both intermediate and final checkpoints.
- 4 downstream benchmarks appropriate for small models with an additional 4 evaluations created from one of the benchmarks translated into 4 different languages
- Perplexity evaluations on 9841 internet domains (e.g. wikipedia.org) represented by a sample of 25 web pages each, derived from Common Crawl.
- Sampling training data from a 100B token corpus, derived from Common Crawl
- Pretraining experiments at the 160M parameter 3.2B token (“chinchilla optimal”) comparing their proposed data recipe to baselines including uniform sampling, language filtering, an n-gram overlap method (DSIR; Xie et al., 2023), and a SOTA  handcrafted FastText classifier (Li et al., 2024)

With this set up they find that:
- Their proposed approach performs comparably to the state of the art handcrafted baseline (which was constructed with expensive pretraining experiments not required by this paper’s approach) and outperforms all other baselines.
- Using Spearman rank correlation as opposed to the authors’ proposed correlation measure performs comparably (figure 6, appendix)
- Ablating the webpage-level filtering using the FastText classifier trained on their selected domains, does lower performance (though the difference is smaller when using Spearman rank correlation).

Their method can also be used to predict the ranking of the 90 public models by downstream performance using only domain perplexities. They find:
- A baseline approach that predicts the downstream rank by a model’s rank w.r.t. to  mean loss over domains already gets R^2 values from 75 to 92 (always within 5 points of the proposed approach)
- Testing over all 8 downstream benchmarks, the proposed approach gets a statistically significantly better R^2 with p=0.035, though no individual benchmark is significantly improved.

They also provide an extensive theoretical statistical framework for using a single-index model (SIM) to represent the relationship between domain perplexity and downstream performance. They explain the use of the empirical CDF rather than raw perplexity values to better handle outliers. They also provide an approach to constrain the correlations to be a valid distribution over domains and to respect the limited number of tokens available per domain. However their empirical results “suggest that the performance of [their] approach is not dependent on the precise form of the estimator that is coupled to [their] theory results, but holds broadly across perplexity-correlation relationships” (line 470).

**Strengths:**

This approach provides a promising way to make use of “the millions of dollars collectively spent” (line 39) on the experimentation represented by all open weight models. The approach extracts information about performant pretraining data selection even when information about the pretraining data of these open-weight models is not available.

Their approach is intuitive and the core of it is quite simple (appearing to be insensitive to the trickier details suggested by theory). This means it should be straightforward for others to apply it.

The authors find that their approach achieves comparable results to a SOTA quality classifier that utilized extensive, costly pretraining experiments, while crucially the authors’ approach requires no new pretraining experiments, unlike other methods..
The paper provides a sound empirical validation of their approach at the 160M parameter scale. They use downstream evaluations that are appropriate for the parameter scale of their experiments (even including some basic multilingual evaluations). When optimizing for a single benchmark, the approach outperforms the state-of-the-art data selection method from DCLM if the latter is handicapped by removing the “manual filtering” step, but not if that manual filtering is included. Their set up demonstrates the utility of their approach for data with a very large number of domains (far greater than can be optimized with brute force pretraining experiments). These results should be more than enough to motivate further exploration of this technique at larger scales.

I appreciated the additional analysis of outliers where the approach doesn’t do so well at predicting relative performance.  The observation that models with “unusual” data mixtures is interesting and will be useful in future work in this space.

**Weaknesses:**

I think the biggest weakness is that the paper’s method targets a single benchmark.  Are there obvious ways to extend the work to target some aggregation of benchmark scores?  Sure, but the paper offers no theory to suggest how we might do that, and no experiments to show whether the obvious ways of doing that would work.
Lambada, which accounts for 5 out of 8 of downstream evaluations, is a language modeling task (i.e. next word prediction) just like the perplexity measures being used as proxies in the proposed approach. This may somewhat obscure whether the proposed approach works as well for all kinds of downstream tasks which typically are not language modeling tasks. That said, the other 3 non-lambada tasks do also benefit from the proposed approach.
It is unclear in the empirical results of the paper what parts of the theoretical results are actually necessary for the evident success of the authors’ approach. The results in Appendix G do not conclusively favor the author’s proposed correlation measure nor their projection method. While the authors do say so themselves (line 467), it would be kinder to the reader to let the reader know this before they have waded through the dense theoretical sections of the paper.
The presentation of the paper also makes it unclear what theoretical results are necessary as the discussion of theory and methods are quite entangled. The paper would be much more readable if there was a simple discussion of what the proposed approach does and then only after that provide theoretical arguments for why this approach is used. I believe Algorithm 1 in the appendix provides the clearest end-to-end description of what exactly the proposed approach is. But even there I am left uncertain of details not mentioned there such as the theta projection or the use of empirical CDF. Is theta projection always used for the greedy data selection, or can the raw gamma values be used without projection since all you need is the order of the domains by the correlation?
I believe the experiments work only at the internet domain granularity for selection.  The paper actually says very little about what a “domain” is – it seems to follow from variables defined in the redpajama data, and there are around 9000 “domains,” so the problem is of interesting size.  But I didn’t come away with a strong sense that – beyond empirical reports – there really should be a relationship between these particular “domains” and model performance on the benchmarks.  The paper would be stronger with some discussion of the qualitative assumptions that lead us to believe perplexity on specific slices of pretraining data (whether defined by internet domains or some other attributes) should be predictive of task performance. Are those assumptions grounded in any prior work?  Are they pervasive?  Are they novel?

The largest model on which the method is tested is only 160M parameters; it’s not clear whether the method will scale.  (I have no reason to believe it won’t, or that it will.)  This scale of model also implies a choice of target benchmarks that deviates pretty far from metrics most builders of language models would care about; I understand that the experiments needed to use benchmarks that would show meaningful differences at the 160M scale, but this creates even more distance between what the paper shows and what is needed in practical settings.

Apart from the most obvious case (based on language ID), the paper really doesn’t delve into what the predictive model itself tells us about the relationship between pretraining data composition and the benchmarks.  This is in part due to the unexciting set of benchmarks, I think.

**Questions:**

1. It seems that factors such as benchmark contamination or formatting overlap between pretraining data and perplexity validation data could alter the relationship between domain perplexity and downstream performance for some observed models. This would introduce more structured noise than simple run-to-run variance due to differing training seeds. In the observed data do correlations vary dramatically for any subsets of models (i.e. are there subsets of models perhaps with systematically biased perplexity to downstream correlations)? Does your approach mitigate this possibility in any way other than by accumulating enough diverse observations that it’s unlikely a substantial number of models will be affected by the same biases?
2. Is the projection method used in algorithm 1 when selecting data or only when predicting downstream performance ranking? Figure 6 implies that the data selection is different when using different projection methods but algorithm 1 doesn’t mention the projection step.
3.  What does “features” refer to on line 344?
4. Line 460: what is meant by “​​significant enrichment of the corresponding languages”
5. Since the domain perplexities are measured over samples of 25 webpages, do the number of tokens used to estimate perplexity vary considerably from domain to domain? This may add noise to your perplexity estimates proportional to how severely some domains are subsetted with respect to tokens as observed by https://arxiv.org/abs/2312.10523 in their subsampling guideline.
6. How are the standard deviations computed for the figure 4 ranking plots?

Other comments (not questions and no response needed; wasn't sure where to put them):

The paper grounds the method in some theory by relating it to estimators for single index models.  This is not extremely important from my point of view, because the method is fairly intuitive, but it may give further intuition or justification to some readers.  The value may be in revealing the assumptions made by the model (through the lens of statistical estimation), so that they can be revised in future explorations.  I thought the discussion of the projections (4.2.3) was mostly unnecessary; I would have rather seen the experimental comparison of the different variants reported in the main paper and the derivations in the appendix, rather than the reverse.

Suggestions (no response needed):

It would be interesting to see more analysis of specifically what domains are selected by the proposed approach contrasted with other baselines. Likewise how do the domains selected differ for different target downstream benchmarks (other than in terms of language)? Also are the most correlated domains also domains with few or many tokens? That is, does the final data selection only include a small subset of the positive correlation domains before satisfying the token budget or are many smaller domains selected?

Line 93: There is some work which uses pretraining experiments to explore high dimensional mixtures of pretraining data (https://arxiv.org/abs/2410.15661).

Line 97: There is work that directly shows the relationship between loss to downstream performance (https://arxiv.org/abs/2403.08540), though this work does not propose a data selection method.

In line 125 and 128 j switches from indexing documents and domains. Maybe use different indices to make this clearer.

On line 208 and 337 “single-page level” was confusing to me until I recalled that you are working with purely webtext and thus documents are webpages. It would be best to define this term before use to avoid possible confusion with other meanings of the word page.

Line 458: avoid use of “significantly” when not discussing statistical significance.

Line 526: “usual” should be “unusual” here, right?

In 4.2.1, I think the epsilons, not x, should be drawn from a standard normal.

Figure 2 doesn’t clearly label methods the way they are referred to in the text.  It would be helpful to clearly mark, for example, the state-of-the-art methods from the DCLM paper as such and “our” methods (those introduced in this paper).

---

> ### Author Response · Authors · 2024-11-22
>
> We really appreciate your suggestions and the exceptional depth of this review. We’ve attempted to address your points about experiment scale and our theory in the response to all reviewers. Below, we respond to your other points:
>
> Why is there not one estimator and projection method that is conclusively favored?
> * After submitting this paper, we found that the other estimator we tested in our paper (spearman) also fits into our theoretical framework - see the response to all reviewers above.
> * To get more decisive evidence in favor of a particular projection method, we have plans to run more extensive experiments. Although we already have a signal from Appendix G that linear projection is more promising than L2.
>
> Can the raw gamma values be used without projection?
> * An interesting suggestion, but we don’t see a way to do this, because at some stage we train a model on a sampling distribution which must be normalized so that the weights are non-negative and sum to 1. Here is a summary of our method: We compute an estimate for the data weight distribution that is optimal according to the single index model. But we can only trust the estimate up to the ranks, meaning that [-1.1,1.9,3.4] could be our estimate, where the real values might be [0.2, 3.9, 4.1]. Our estimate is also not a proper sampling distribution, so we must project it so that the weights sum to 1 and are non-negative. Then we can use the sampling distribution for pretraining immediately, or train a fasttext classifier on it so that we can generalize our approach to new domains or even apply it at the page-level.
>
> Does it only work with internet-domain-level granularity?
> * We haven’t tested this, but it is a great idea to do so as part of subsequent research in this area.
>
> Is it reasonable to assume that selecting certain domains would lead to better results?
> * It seems reasonable based on our results, and this is what many other quality filtering methods do at some level. For example, the leading filter (the fasttext oh eli5 model) is trained to distinguish certain “domains” (like subsets of reddit) from other “domains” (common crawl).
>
> Can we delve further into the composition of what is selected beyond language ID?
> * The issue here is that we don’t have annotations beyond language ID in RedPajama V2. Would you be supportive if we qualitatively labeled the smaller clusters within the language clusters, in our t-SNE plot? There are clusters that look like they correspond to things like funeral homes, legal services, etc. It seems a bit ad-hoc to perform an analysis in this way, which is why we didn’t do it, but we did find these labels illuminating ourselves.
>
> Do bits-per-byte correlations vary dramatically for any particular families of models across different samples from the pretraining data pool? Can we mitigate it?
> * We haven’t done an experiment examining this. If you have suggestions for how this can be mitigated, if it is an issue, then those would be most welcome!
>
> Does algorithm 1 use the projection step?
> * It does use the projection step. It turns out that the linear projection ends up being a very simple procedure that says “include all of this domain” or “don’t include any of this domain”. This is done in the “Select most to least correlated domains” loop, starting on line 767.
>
> What does “features” refer to on line 344?
> * It refers to the domains. The domains are the features for our single index model.
>
> Line 460: what is meant by “​​significant enrichment of the corresponding languages”?
> * We mean that when the target task is in a particular language, perplexity correlations picks out significantly more tokens in that language compared to other targets.
>
> Since the domain perplexities are measured over samples of 25 webpages, do the number of tokens used to estimate perplexity vary considerably from domain to domain? This might add noise.
> * We haven’t done an experiment examining this, although it is in the queue now behind all of our pressing pretraining experiments.
>
> How are the standard deviations computed for the figure 4 ranking plots?
> * These are computed by applying the formula for standard deviation on the difference between actual and predicted values for each point.

---

> > ### Comment · Reviewer_2JKP · 2024-11-25
> >
> > Thanks for your detailed responses!
> >
> > I want to reiterate a point that was admittedly a bit buried in my review:  "The presentation of the paper also makes it unclear what theoretical results are necessary as the discussion of theory and methods are quite entangled."  I think most readers will find the theory less intuitive and less important than the simple idea of sampling domains greedily in the order of correlation to downstream performance. There may be an audience that is also interested and capable of making use of the further theoretical insights you add on top of this already intuitive approach, but in my view that should be presented as additional information after the simple intuitive description of the approach, clearly separating the “what” and the “why” of what you do. Not doing this limits the impact of the paper by narrowing its audience unnecessarily.
> >
> > To give a small example of what this could look like: Part of “what” the algorithm does is greedly sampling domains in order of their correlation with a downstream task. One way to motivate “why” to do this is that this greedy sampling can be interpreted as a projection of the estimated correlations into a sampling distribution. This would avoid confusion; it wasn't totally clear that these were the same thing until I read your response.
> >
> > Re. "Do bits-per-byte correlations vary dramatically for any particular families of models across different samples from the pretraining data pool? Can we mitigate it?":  a suggestion is to simply to visualize the observational results as plots where the y axis is one downstream task and x axis is a domain perplexity and points are observed models. Are there clusters of models that seem to follow a different trend, or do all models more or less follow the trend besides noise?
> >
> > Re. "Can we delve further into the composition of what is selected beyond language ID?": you ask if qualitatively labeling the clusters would help. I'm inclined to agree that this is mildly interesting but a bit ad-hoc. Instead I would suggest that you analyze the internet domains themselves (i.e. the name: cnn.com or so on). What are the top or bottom k domains most or least correlated with certain domains? Are these well known, large domains, or obscure small domains? Does the selected distribution of domains include many domains or is it dominated by a few? Are steeply does correlation between domains and downstream tasks drop off when sampling in order of correlation? These sorts of analyses might shed more light on what kinds of domains these are and how they are relating to downstream tasks.
> >
> > One remaining question: does algorithm 1 use the empirical CDF of x as in equation 6 or does it not use it as in unnumbered equation on line 180. The Algorithm 1 pseudocode implies it uses the raw rank of x as in the line 180 equation as it takes rank(x_0...x_n) . Clarity about this key detail of the approach would greatly improve impact for readers who will want to adopt your approach.

---

> > > ### Author Response · Authors · 2024-11-26
> > >
> > > Thanks for these great followup suggestions. Here’s a response to them and also a response to your question:
> > >
> > > We could make the writing more intuitive, first presenting the basic idea of selecting greedily by correlations and then presenting the theory
> > > * This is a good idea. We had one round of rewriting before we submitted to ICLR that was targeted at this. It looks like it could benefit another round, and so we will keep this in mind for the next version of our paper.
> > >
> > > Bits-per-byte model variance suggestion
> > > * This is a good idea too. It could be a more direct way of getting at what is going on with the outlier models in Figure 4. We will explore this.
> > >
> > > Composition of selected domains:
> > > * The top most correlated domains per task should be straightforward to add to the appendix. The most correlated domains for a task tend to look somewhat interpretable. For example, book domains are highly selected for LAMBADA, which makes sense because the LAMBADA benchmark comes from book excerpts.
> > >
> > > Does algorithm 1 use the empirical CDF?
> > > * Yes, it uses the empirical CDF implicitly. Using the empirical CDF is the same as using the ranks here (the vector of ranks and the vector of empirical CDFs just differs by a global constant scaling factor which does not affect the projected values).

---

### Official Review · Reviewer_Pj1w · 2024-11-04

**Soundness:** 3
**Presentation:** 2
**Contribution:** 2
**Rating:** 5
**Confidence:** 3

**Summary:**

This paper analyzes how to select high-quality pretraining data for large language models (LLMs) without training new LLMs. The authors find that LLM losses on certain pretraining texts correlate with downstream benchmark performance. The key idea is to select data domains where lower loss generally corresponds to higher downstream performance.
To do this, the authors use a collection of pre-trained LLMs and measure the log-likelihoods for texts from various domains and the performance of these LLMs on downstream benchmarks. They then calculate the correlation between these two measures for each domain. Domains with high correlations are selected for pretraining.

**Strengths:**

The paper presents a new and remarkably simple approach to data selection for LLM pretraining. It leverages readily available resources (publicly available LLMs) and a straightforward correlation-based method.  The authors ground their approach in a statistical framework, providing a theoretical basis for their correlation-based data selection. The authors demonstrate the robustness of their approach by showing its effectiveness across various benchmarks and conditions.

**Weaknesses:**

While the paper presents promising results at the 160M parameter scale, further validation at larger scales is necessary to assess the scalability and effectiveness of the approach for slightly more massive LLMs. The method inherits the biases present in the publicly available LLMs used for estimating correlations. This raises potential concerns about bias amplification, especially if these LLMs were trained on data with inherent biases.

**Questions:**

- Relevant works that are not cited:

[1] Niklas Muennighoff, Alexander M. Rush, Boaz Barak, Teven Le Scao, Aleksandra Piktus, Nouamane
Tazi, Sampo Pyysalo, Thomas Wolf, and Colin Raffel. Scaling data-constrained language models, 2023

[2] Max Marion, Ahmet Üstün, Luiza Pozzobon, Alex Wang, Marzieh Fadaee, Sara Hooker. When Less is More: Investigating Data Pruning for Pretraining LLMs at Scale, 2023

---

> ### Author Response · Authors · 2024-11-22
>
> Thanks for raising these points about experiment scale - see our response above to all reviewers for our attempt to address them. Regarding the concerns about amplifying bias of existing models, we are curious what specific issues you see? Is this an ethics concern, or are you using the term “bias” to mean something more general? Also, thank you for providing some relevant papers which we missed - we will cite them!

---

> > ### Author Response · Authors · 2024-12-03
> > **Thoughts on our new experimental results?**
> >
> > Thank you again for the feedback - we ran experiments at ~ 2-3x larger scales, and the results remain positive. We are wondering if you have a chance to take a look and have any responses?

---

### Official Review · Reviewer_54hW · 2024-11-04

**Soundness:** 3
**Presentation:** 2
**Contribution:** 2
**Rating:** 6
**Confidence:** 4

**Summary:**

This paper proposes a new LLM pre-training data selection framework where no LLM training is needed. Their framework leverages publicly accessible pre-trained LLMs and is based on the correlation between downstream benchmark performance and LLMs' losses. In the paper, the authors provide theoretical background for the proposed framework and present experiments where they use 160M parameters models with a 3.2B token budget, comparing DSIR and DataComp-LM data selection. Concretely, they selected data from RedPajamas V2, and used LAMBADA, ARC-E, PIQA, and SciQ as target downstream benchmarks. In their experiments, the proposed method outperforms DSIR and shows competitive performance with DataComp-LM best data selection method.

**Strengths:**

1. This paper focuses on avoiding costly pre-training ablations for LLM pre-training data selection and proposes a method that leverages public, pre-trained LLMs based on the correlation between downstream performance and perplexity. This aim is quite important for today's LLM practitioners and the proposed framework is very original.

2. Although limited (see below "weaknesses"), the proposed framework demonstrates promising results compared to baselines.

3. Although the presentation can be improved, the paper includes a theoretical background for the proposed framework.

**Weaknesses:**

1. The main weakness of this paper is the scale of the experiments. The paper only includes pre-training experiments using 160M parameters LLM with a 3.2B token budget. This is very small and creates questions about the validity of the results. For example, the smallest scale for DataComp-LM paper is 400M and 8.2B and they experimented with 5 different scales which goes up to 7B parameters model trained on 276B tokens (as DCLM competition scale).

2.  Experiments only include 8 evaluation benchmarks (5 of them are Lambada from different languages). This is a main limitation for both showcasing the validity of results but also limit the understanding how the proposed framework. work when there are many more benchmarks to consider for perplexity validation. Again for comparison, DataComp-LM includes an extended evaluation suite that consists of up to 53 downstream tasks.

3. If the framework only optimizes data selection per task, is it not expected to have good scores on optimized tasks? Are there any experiments where authors tested aggregating benchmark scores as one overall metric?

**Questions:**

1. Clarification for Line 201: `select domain greatest to least \gamma, taking all the tokens in each domain once, until we exhaust our total pretraining token budget.`
Does this mean that if a domain with the greatest \gamma includes more tokens than the token budget, only tokens from that domain will be selected?

2. For the proposed perplexity correlation, have you tested aggregating benchmark scores in some way to use as an overall benchmark metric? From Figure 2, I only see LLMs where data selection (ppl correlations) is optimized for each target task.

---

> ### Author Response · Authors · 2024-11-22
>
> Thanks for bringing up these points about experiment scale and about how our approach works with an aggregation of benchmarks - see our response above to all reviewers where we try to address these concerns. Regarding your question about a domain with more tokens than the budget: you are right that if a domain has more tokens than the pretraining budget, our estimator will only select tokens from that domain. It is unclear what happens experimentally in that case - in practice, the number of tokens from even the largest domains tends to be miniscule relative to total pretraining budgets. This degeneracy is also resolved by training a fasttext model and then applying the model as a pretraining data filter instead of using the estimate directly.

---

> > ### Comment · Reviewer_54hW · 2024-11-25
> >
> > Thanks for your response and for running experiments at the 410M parameter scale. Could you please add the results to the paper or the rebuttal? The author's comment on finding evaluation tasks where the model performs above random performance is reasonable. Since they conducted additional experiments and based on their response, I have updated my score accordingly.

---

> > > ### Author Response · Authors · 2024-11-26
> > >
> > > Thanks - we have included the figure for the 410M model experiments in the supplementary materials, so you can see it now. We plan to write about the 410M experiments and add the figure to next version of the paper, along with the other revisions that we will make.

---

### Author Response · Authors · 2024-11-22
**Response to All Reviewers**

We appreciate this thoughtful and detailed set of reviews. In this response to all reviewers, we address their shared concerns. Separately, we address specific reviewer concerns in individual messages below the reviews.

The scale of the pretraining experiments is small (all reviewers):

* We have duplicated some of our original pretraining experiments at the 410M scale instead of the 160M scale, and are currently running them. Due to the time constraints for our rebuttal, we are only running the perplexity-correlations models and the uniform-sampling baseline model. To make the task harder for perplexity-correlations (and in the interest of time), we are using the same 3.2B datasets computed by perplexity correlations for the 160M parameter models, and duplicating them to be 8.2B tokens, which is chinchilla optimal at the 410M scale. The uniform-sampling model selects 8.2B tokens from the pre-filtered pool, meaning that it sees many more unique tokens than the perplexity-correlations models. If perplexity-correlations didn’t work at this scale, we would expect the target models to perform worse (not even the same) as the uniform-sampling model. We already have results for the SciQ target, the Arc Easy target, the PIQA target, and the uniform baseline. In these cases so far, our target models score well over a point higher on the respective benchmarks than the uniform baseline, except for PIQA which gets about the same (0.2% difference). We will add the 410M suite of experiments to the next version of our paper.
* EDIT: all 410M models have finished for this experiment, and perplexity-correlations does better than uniform sampling at all of our 8 tasks by a decent margin (~one percent to a few percent), except for PIQA where it is about the same (0.2% difference).

Our experiments only target individual benchmarks, not an aggregate (reviewers 54hW, 2JKP, 9vdE):

* Although we would hope that increasing performance for one benchmark would increase performance on many others (https://x.com/gblazex/status/1746295870792847562), we see no theoretical issue with using perplexity correlations to target an aggregate of benchmarks. This aggregation would itself be another benchmark. We have rented an H100 node and are currently running pretraining experiments where we target the DataComp-LM “core” benchmark, which is an aggregate of 22 benchmarks. This is particularly challenging because many of these 22 benchmarks yield random performance at the 160M and even 410M scales. We are currently training 160M, 410M, and 1B models with chinchilla optimal token counts. Although we may not be able to provide final numbers before the rebuttal deadline, we will put DataComp-LM numbers in the next version of our paper, regardless of whether they are good or bad.
* By choosing DataComp-LM core, we also hope to address a concern raised by reviewer 54hW about the limited number of benchmarks we tested so far.

The theory did not explain why another estimator (Spearman) also does well (reviewers 2JKP, 9vdE):

* We have now proven that the Spearman estimator is also monotonic with respect to the optimal weights in expectation, so it fits into our theory. We will update the next version of the paper with this proof. We can see from the common formula for Spearman rank correlation (where there are no ties between points) that we can reduce the problem to finding $E[(\Phi(y_i) - \Phi_j(x_{i,j}))^2]$. We can just multiply this out and use linearity of expectation to get $E[\Phi(y_i)^2] - 2E[\Phi(y_i) \cdot \Phi_j(x_{i,j})] + E[\Phi_j(x_{i,j})^2]$. Because $x$ and $y$ are both gaussian under the high dimensional regression assumptions, we can compute most of the terms easily. The middle term $2E[\Phi(y_i) \cdot \Phi_j(x_{i,j})]$ takes a bit more work, but we can use the law of total expectation and prove an analogous lemma to lemma 1. Then we can continue to proceed in a way analogous to the original estimator proof in our paper. Eventually we get to an integral that we found in a table.

We do not provide experiments where we train our estimator on smaller models and extrapolate to larger models (reviewers 9vdE, TyHg):

* We agree that such an experiment would be ideal to run. It would be exciting to use our approach to gain information about which sampling distributions are useful for next-gen models which would likely be larger than current-gen models. We plan on another experiment in a future version of our paper where we use only models on the order of 100M parameters for our estimate and train a model on the order of 1B parameters. Due to compute constraints, we may have to choose only a few small scale benchmarks out of the 8 that we used in our paper. Alternatively, if any of the reviewers have suggestions for any benchmarks on which ~100M parameter models get significantly above random, we would be open to considering them too.

---

### Meta-Review · Area_Chair_ufNp · 2024-12-20

**Metareview:**

The paper proposed a novel framework for selecting pretraining data based on correlations between LLM's perplexity and downstream benchmark performance. The authors leverage publicly available LLMs, their perplexity on different domain, their performance on downstream tasks, to estimate these correlations (within in a statistical framework), thereby avoiding costly pretraining ablations. Experiments at the 160M parameter scale with a 3.2B token budget, shows that the framework is superior to existing baselines.

The proposed approach is theoretically grounded, computationally efficient, and inspiring. It achieves promising results in data selection, but the paper’s contribution is relatively limited by the small scale of experiments, especially considering 3.2B token budget is much smaller than real training of LLMs.

The paper is a well-designed academic work and also has the potential to become a practical framework for pretraining data selection.  Based on the reviewers' scores, the positive trajectory of the rebuttal, the submission is recommended for acceptance.

**Additional Comments On Reviewer Discussion:**

Major concern from several reviewers is the scale of experiments. The author responded with new experiments on 410M model and 8.2B tokens, and reported consistent improvements across benchmarks.

The limited number of benchmarks are raised by reviewer 54hW, 9vdE. Authors promised to do experiments on more benchmarks.

---

### Decision · Program_Chairs · 2025-01-22

Accept (Poster)